# Phenotypic and Genotypic Characterization of ESBL-, AmpC-, and Carbapenemase-Producing *Klebsiella pneumoniae* and High-Risk *Escherichia coli* CC131, with the First Report of ST1193 as a Causative Agent of Urinary Tract Infections in Human Patients in Algeria

**DOI:** 10.3390/antibiotics14050485

**Published:** 2025-05-09

**Authors:** Hajer Ziadi, Fadela Chougrani, Abderrahim Cheriguene, Leticia Carballeira, Vanesa García, Azucena Mora

**Affiliations:** 1Laboratorio de Referencia de Escherichia coli (LREC), Departamento de Microbioloxía e Parasitoloxía, Universidade de Santiago de Compostela (USC), 27002 Lugo, Spain or hajer.ziadi.etu@univ-mosta.dz (H.Z.); leticia.carballeira.campa@usc.es (L.C.); 2Laboratory of Animal Production Science and Technology, Faculty of Natural and Life Sciences, Abdelhamid Ibn Badis University, Mostaganem 27000, Algeria; chougranifadela@yahoo.fr; 3Biodiversity Laboratory, Water and Soil Conservation, Mostaganem 27000, Algeria; 4Laboratory of Bioeconomy, Food Security and Health, Faculty of Natural and Life Sciences, Abdelhamid Ibn Badis University, Mostaganem 27000, Algeria; abderrahim.cheriguene@univ-mosta.dz; 5Instituto de Investigación Sanitaria de Santiago de Compostela (IDIS), 15706 Santiago de Compostela, Spain; 6iARCUS Aquatic One Health Research Center, Universidade de Santiago de Compostela, 15705 Santiago de Compostela, Spain

**Keywords:** urinary tract infection (UTI), *Escherichia coli*, *Klebsiella pneumoniae*, ST131, ST1193, ESBL, carbapenemases, Algeria

## Abstract

Background: High-risk *Escherichia coli* clones, such as sequence type (ST)131 and ST1193, along with multidrug-resistant (MDR) *Klebsiella pneumoniae*, are globally recognized for their significant role in urinary tract infections (UTIs). This study aimed to provide an overview of the virulence factors, clonal diversity, and antibiotic resistance profiles of extended-spectrum cephalosporin (ESC)-*E. coli* and *K. pneumoniae* causing UTIs in humans in the Tebessa region of Algeria. Methods: Forty *E. coli* and 17 *K. pneumoniae* isolates exhibiting ESC-resistance were recovered (July 2022–January 2024) from urine samples of patients at three healthcare facilities to be phenotypically and genotypically characterized. Whole genome sequencing (WGS) was performed on the ST1193 clone. Results: Among *K. pneumoniae* isolates, all except one harbored CTX-M-15, with a single isolate carrying *bla*_CTX-M-194_. Additionally, two *K. pneumoniae* isolates co-harboring *bla*_CTX-M-15_ and *bla*_NDM_ exhibited phenotypic and genotypic hypervirulence traits. Fluoroquinolone resistance (FQR) was detected in 94.1% of *K. pneumoniae* isolates. The *E. coli* isolates carried diverse ESC-resistance genes, including CTX-M-15 (87.5%), CTX-M-27 (5%), CTX-M-1, CMY-59, and CMY-166 (2.5% each). Co-carriage of *bla*_ESC_ and *bla*_OXA-48_ was identified in three *E. coli* isolates, while 62.5% exhibited FQR. Phylogenetic analysis revealed that 52.5% of *E. coli* belonged to phylogroup B2, including the high-risk clonal complex (CC)131 CH40-30 (17 isolates) and ST1193 (one isolate). In silico analysis of the ST1193 genome determined O75:H5-B2 (CH14-64), and the carriage of IncI1-I(Alpha) and IncF [F-:A1:B10] plasmids. Notably, core genome single-nucleotide polymorphism (SNP) analysis demonstrated high similarity between the Algerian ST1193 isolate and a previously annotated genome from a hospital in Northwest Spain. Conclusions: This study highlights the spread and genetic diversity of *E. coli* CC131 CH40-30 and hypervirulent *K. pneumoniae* clones in Algeria. It represents the first report of a CTX-M-15-carrying *E. coli* ST1193 in the region. The findings emphasize the urgent need for antibiotic optimization programs and enhanced surveillance to curb the dissemination of high-risk clones that pose an increasing public health threat in Algeria. A simplified method based on virulence traits for *E. coli* and *K. pneumoniae* is proposed here for antimicrobial resistance (AMR) monitoring.

## 1. Introduction

Urinary tract infections (UTIs) are the second most common infectious disease in humans, after respiratory tract infections, in healthcare-associated centers and the community worldwide [1]. UTIs affect millions of individuals annually, are associated with decreased quality of life, and pose a significant clinical and economic burden [2]. Women are particularly susceptible, with approximately 50% to 60% of women experiencing at least one UTI during their lifetime [3,4]. Data from 1990 to 2021 revealed that UTIs showed an upward trend, with a particularly pronounced disease burden among women, older men, and low–middle sociodemographic index regions [5]. Furthermore, the impact estimation of UTIs in 2021 was 287,200 deaths associated with bacterial antimicrobial resistance (AMR), including 67,467 deaths directly attributable to bacterial AMR, being the fourth leading cause of death associated with AMR worldwide [5,6]. Tropical Latin America and low–middle sociodemographic index regions exhibited the highest impact in terms of prevalence, death rate, and disability-adjusted life-years (DALYs) [5].

A wide range of virulence factors and multidrug-resistant (MDR) pathogenic strains are described in the pathogenicity and resistance of the uropathogenic agents, making it more difficult to manage these complicated infections. Among them, *Escherichia coli* is the predominant causative agent of UTIs, and *Klebsiella pneumoniae* is recognized as the second contributor to UTIs. Both *E. coli* and *K. pneumoniae* exhibit a variety of virulence factors that enable them to adhere to, invade, and persist in the urinary tract, often resulting in acute or chronic infections [4]. These pathogens also exhibit increasing resistance, especially to cephalosporins, fluoroquinolones, and carbapenems, leading to longer hospital stays, increased healthcare costs, and treatment failures [6]. Globally, 0.26 million deaths (95% uncertainty interval [UI]: 0.18–0.36) were associated with bacterial AMR in UTIs in 2019 [7].

Among *E. coli* strains, infections of MDR high-risk clones such as ST38, ST131, ST167, ST405, ST410, ST648, and ST1193 are commonly reported [8,9,10]. ST131 is considered the most prominent, while ST1193 appears to be emerging as a significant clone following a trajectory similar to that of the globally dominant ST131 lineage. It exhibits strong virulence potential, efficient human-to-human transmission, and adaptability to both community and hospital environments. Its rise is particularly concerning due to its co-resistance to multiple antibiotic classes and its frequent carriage of key resistance determinants such as *bla*_CTX-M_ and mutations in *gyrA*/*parC*, conferring fluoroquinolone resistance (FQR) [11]. MDR clones are often linked to resistance to FQR, with many also producing cefotaximases (CTX-M), a family of extended-spectrum beta-lactamase (ESBL) enzymes which confer resistance to extended-spectrum cephalosporins (ESC), thereby limiting the effectiveness of this last resort class of antibiotics and significantly reducing available treatment options in clinical settings [9,10,11,12].

On the other hand, ESBL and carbapenemase-encoding *K. pneumoniae* are globally disseminated and cause infections that are often difficult to treat. According to the 2024 WHO Medically Important Antimicrobials (MIA) List, ESBL-producing and carbapenemase-producing *E. coli* and *K. pneumoniae* are highlighted as critical threats due to their resistance to critically important antimicrobials (CIAs), underscoring the need for strengthened antimicrobial stewardship across human, animal, and environmental health sectors [13].

In Algeria and other low- and middle-income countries (LMICs), the prevalence of MDR uropathogens is notably high, exacerbated by the lack of robust antibiotic stewardship programs. The study “Global burden of bacterial antimicrobial resistance in 2019: a systematic analysis”, published in The Lancet [6], reported that the Middle East and North Africa (MENA) region experienced a significant impact from AMR in 2019. Specifically, there were approximately 7687 deaths (95% uncertainty interval: 6663–8969) associated with UTIs in the MENA region during that year.

Recent studies have underscored the growing public health threat posed by MDR uropathogens in Algeria, particularly *E. coli* and *K. pneumoniae*. In a study conducted in the Tizi-Ouzou region between 2017 and 2019, *E. coli* isolates recovered from UTI patients exhibited high levels of resistance to frequently used antibiotics, including ampicillin, trimethoprim-sulfamethoxazole, and fluoroquinolones [14]. In parallel, several investigations have reported the emergence and dissemination of MDR *K. pneumoniae* in Algeria. Notably, carbapenemase-producing strains such as KPC-2 and OXA-48 have been detected in clinical settings, with clonal dissemination of high-risk lineages like ST258, ST101, and ST512 [15,16]. For instance, KPC-2-producing *K. pneumoniae* ST512 was isolated from cerebrospinal fluid in 2014 [15], while a 2016–2017 study in Annaba Hospital documented the first cases of KPC-2-producing *K. pneumoniae* ST258, alongside an outbreak involving ST101 strains harbouring the *bla*_OXA-48_ gene in a urology department, suggesting nosocomial transmission [16]. Moreover, isolates collected in 2018 from a military hospital in Oran showed carriage of ESBL-encoding genes such as *bla*_CTX-M_ and *bla*_TEM_, often associated with biofilm formation, which can further complicate treatment and infection control [17]. These data collectively highlight the alarming spread of MDR Enterobacteriaceae and the importance of coordinated national strategies to curb resistance in both community and hospital settings. In Tebessa province, another study demonstrated the correlation between inappropriate antibiotic use and the increasing prevalence of ESBL-producing bacteria [18]. Similar patterns have been observed in other regions, where antibiotics are frequently dispensed without prescriptions or proper medical oversight [19,20]. These challenges are compounded by the limited availability of coordinated national surveillance data, with most reports stemming from academic studies rather than systematic public health monitoring [21].

This study aims to contribute to this growing body of evidence by characterizing the virulence and antibiotic resistance profiles of *E. coli* and *K. pneumoniae* isolates causing UTIs in the Tebessa region of Algeria. By exploring a simplified laboratory workflow for AMR monitoring, this work seeks to support the development of cost-effective surveillance approaches. A clearer understanding of local epidemiological trends is essential to guide therapeutic choices, inform infection control measures, and strengthen national AMR surveillance capacity.

## 2. Results

The present study comprised 57 non-duplicate isolates, 40 *E. coli* and 17 *K. pneumoniae*, recovered from urine samples at three healthcare facilities in northeast Algeria from July 2022 to January 2024 (Appendix A).

### 2.1. Escherichia coli Collection

A total of 40 non-duplicate *E. coli* isolates were recovered from patients in three healthcare facilities in the Tebessa region, Algeria, including 9 isolates from male and 31 from female patients. All isolates tested positive for the *uidA* target-specific species gene [22]. Phenotypic characterization on MacConkey Lactose agar revealed that four isolates (10%) were non-lactose fermenters (NLF) (Appendix A).

#### 2.1.1. Antimicrobial Susceptibility Testing (AST) and Genotypic Characterization of *bla* Genes

Antimicrobial susceptibility testing (AST) revealed that all *E. coli* isolates were resistant to ampicillin, cefuroxime, cefotaxime, and ceftazidime. As shown in Figure 1, most of the isolates (≥50%) exhibited resistance to aztreonam, amoxicillin-clavulanic acid, nalidixic acid, sulfamethoxazole, ciprofloxacin, doxycycline, and tobramycin. Notably, four isolates (10%) were resistant to ertapenem. All isolates were classified as MDR, exhibiting resistance to at least one drug in three or more antimicrobial categories [23], and 62.5% demonstrated FQR (Figure 1).

The isolates were subsequently analysed by PCR to detect the presence of *bla*_ESBL_ (SHV, CTX-M), *bla*_CMY_, and *bla*_CARBA_ (VIM, IMP, OXA, NDM, KPC) genes, followed by Sanger sequencing. The collection exhibited different ESC enzymes, including CTX-M-15 (87.5%), CTX-M-27 (5%), as well as CTX-M-1, CMY-59, and CMY-166 (2.5% each). The *bla*_OXA-48_ gene was detected in three *E. coli* isolates resistant to ertapenem, which also harboured CTX-M-27 (two isolates) and CMY-59 (one isolate). None of the isolates tested positive for *mcr* (1 to 5) genes or showed polymyxin resistance (Appendix A).

#### 2.1.2. Virulence Traits

The screening of virulence factors associated with enhanced urinary tract colonization indicated that 18 out of the 40 isolates (45%) met the criteria for UPEC status, defined by the presence of at least three of the specific virulence genes (*chuA*, *fyuA*, *vat*, and *yfcV*) [24]. Individually, *chuA*, *fyuA*, *vat*, and *yfcV* were detected in 77.5%, 80%, 5%, and 52.5% of the isolates, respectively.

Furthermore, 72.5% of the isolates were classified as ExPEC based on the presence of at least two of the five key virulence markers: *papAH*, *sfa/focDE*, *afa/draBC*, *kpsMII*, and *iutA* [25]. The individual prevalence of these genes among the isolates was 40%, 0%, 30%, 57.5%, and 77.5%, respectively (Appendix A).

#### 2.1.3. Clonal Groups

Using the PCR method described by Clermont et al. [26,27], five *E. coli* phylogroups were identified among the 40 isolates. Phylogroups B2 and D were the most prevalent, accounting for 52.5% and 22.5% of the isolates, respectively. The remaining isolates belonged to phylogroups B1 (12.5%), A (7.5%), and F (5%) (Table 1).

Presumptive identification of the pandemic clonal complex (CC)131 was performed via PCR screening for the *rfb*O25b and *fli*C_H4_ gene markers [28,29] with 17 out of 40 isolates (42.5%) testing positive for both. Additionally, the *fli*C_H5_ gene, typically associated with clones such as ST1193 or CC131 of clade A [9,30] was determined in one NLF isolate (2.5%) (Appendix A).

Clonotyping identified 14 distinct *fumC-fimH* combinations. Notably, 42.5% of the isolates exhibited clonotype CH40-30, which is typically associated with CC131 [9,31,32]. The second most prevalent clonotype, CH26-5, was detected in 15% of the isolates. Notably, UPEC status was associated with phylogroup B2 isolates, whereas ExPEC status was linked to both B2 and D isolates (Table 1).

**Table 1 antibiotics-14-00485-t001:** Clonotypes determined among the 40 *E. coli* isolates.

Phylogroup	Clonotype (CH) ^a^	No Isolates	UPEC Status ^b^	ExPEC Status ^c^	FQR
B2	CH40-30	17	15	16	17
	CH14-64	1	1	1	1
	CH108-Neg	2	2	2	1
	CH1012-Neg	1	0	0	1
D	CH26-5	6	0	6	1
	CH26-Neg	3	0	3	0
B1	CH65-27	2	0	0	0
	CH27-54	1	0	0	0
	CH65-32	1	0	0	0
	CH7-604	1	0	0	0
A	CH11-54	1	0	1	1
	CH11-Neg	1	0	0	1
	CH7-94	1	0	0	0
F	CH88-Neg	2	0	0	2

^a^ Eight distinct *fimH* alleles were determined within the 40 *E. coli*. (Neg): Nine isolates showed no amplification of the 489-bp internal sequence of the *fimH* gene targeted for clonotyping [31] despite all 40 *E. coli* isolates testing positive for *fimH* using primers and conditions described by Johnson and Stell [33]. ^b^ UPEC status. + positive for ≥3 of the following genes: *chuA*, *fyuA*, *vat*, and *yfcV* [24]. ^c^ ExPEC status. + positive for ≥2 of these five markers: *papAH*, *sfa/focDE*, *afa/draBC*, *kpsMII,* and *iutA* [25]. FQR, fluoroquinolone resistance.

##### Characterization of Clonal Complex (CC) 131 Isolates

The most prevalent clonal group O25b:H4-B2 (CH40-30) CC131, identified in 17 isolates, exhibited FQR (100%), carriage of *bla*_CTX-M-15_ (100%), and virulence profiles accomplishing ExPEC (16 of 17, 94.1%) and UPEC (15 of 17, 88.2%) status (Table 1). Remarkably, one CC131 isolate displayed non-susceptibility to ertapenem, though the underlying genetic mechanism could not be determined. High levels of resistance were observed against tobramycin (16 isolates, 94.1%), amoxicillin/clavulanic acid (15 isolates, 88.2%), sulfamethoxazole (13 isolates, 76.5%), gentamycin (13 isolates, 76.5%), and doxycycline (10 isolates, 58.8%).

Using the virotyping scheme proposed by Dahbi et al. [30], virotype F was identified in seven isolates (41.2%), while virotype E was detected in three isolates (17.6%). Among the extraintestinal virulence markers included in this scheme, the specific pilus tip adhesin molecule type II associated with pyelonephritis (*papGII*) was present in all isolates, the secreted autotransporter toxin (*sat*) was detected in 14 isolates (82.3%), and the K5 group II capsule (*kpsMII*-K5) was found in 10 isolates (58.8%). Notably, the co-occurrence of *pap*GII, *cnf*1 (cytotoxic necrotizing factor 1), and *hly*A (α-hemolysin) was observed in three isolates of virotype E(17.6%) (Table 2).

##### In Silico Characterization of ST1193 Isolate

The NLF isolate B2-CH14-64 underwent WGS and in silico analysis using bioinformatics tools from the Center for Genomic Epidemiology. Table 3 summarizes its genomic characteristics.

Serotyping analysis using SerotypeFinder identified the O75:H5 serotype while CHTyper confirmed its clonotype as CH14-64. Multilocus sequence typing (MLST) based on the seven-gene of the Atchman scheme assigned the sequence type (ST)1193, while the alternative MLST scheme using eight genes (*dinB*, *icdA*, *pabB*, *polB*, *putP*, *trpA*, *trpB*, and *uidA*) [34] predicted ST53. Core genome MLST (cgMLST) analysis, based on 2513 loci, assigned the isolate to cgST140226.

Resistance gene profiling using ResFinder confirmed the presence of *bla*_CTX-M-15_, along with chromosomal mutations associated with FQR (*gyrA* p.S83L, *gyrA* p.D87N, *parC* p.S80I, and *parE* p.L416F), consistent with the observed phenotypic resistance. Additionally, acquired resistance genes, such as *aac(3)-IIa*, *aac(6′)-Ib-cr*, *bla*_OXA-1_, and *catB3* were predicted.

Virulence profiling using VirulenceFinder identified different virulence factors, including *vat*, *fyuA*, *yfcV*, *chuA*, *iutA*, and *kpsM*II, confirming its UPEC and ExPEC status. Plasmid analysis revealed the presence of IncI1-I(Alpha) and IncF [F-:A1:B10] plasmids, along with small Col-like plasmids.

To investigate the genomic relationship between the Algerian ST1193 isolate and five ST1193 isolates recently recovered from a hospital in Northwest Spain, we performed a single-nucleotide polymorphism (SNP) comparison of their core genomes, which represented 95.65% of the reference genome LREC-269 (5.4 Mb). The phylogenetic analysis was conducted using CSI Phylogeny 1.4 (Figure 2A; Appendix A). The Algerian ST1193 isolate (LREC-468) exhibited a minimum pairwise distance of 78 SNPs from LREC-269 and a maximum distance of 128 SNPs from LREC-270.

Additionally, we accessed Enterobase (https://enterobase.warwick.ac.uk/; accessed on 20 December 2024) to search for ST1193 genomes assigned to cgST140226. We identified and retrieved one publicly available genome, ESC_RA5887AA, from BioProject PRJEB21277 registered in 2020 by the University of Oxford. A subsequent SNP comparison (Figure 2B, Appendix A), which included the Algerian (LREC-468), five Spanish ST1193 genomes, and ESC_RA5887AA, showed that 82.6% of nucleotide positions in the reference genome (LREC-269) were conserved across all analysed genomes. The Algerian LREC-468 genome differed by 55 SNPs from ESC_RA5887AA.

### 2.2. Klebsiella pneumoniae Collection

Seventeen *K. pneumoniae* isolates were recovered from patients across three healthcare facilities in the Tebessa region of Algeria, comprising five isolates from males and 12 from female patients. Sixteen isolates tested positive for the *kp50233* species-specific gene. The single negative isolate was confirmed as *K. pneumoniae* using the matrix-assisted laser desorption/ionization-time-of-flight mass spectrometry (MALDI-TOF).

#### 2.2.1. AST and Genotypic Characterization of *bla* Genes

All isolates exhibited resistance to ampicillin, cefuroxime, cefotaxime, and aztreonam. Additionally, the majority (≥60%) were resistant to ceftazidime, ciprofloxacin, fosfomycin, sulfamethoxazole, and amoxicillin-clavulanic acid. Notably, two isolates (11.8%) demonstrated resistance to ertapenem (Figure 3). All isolates were classified as MDR, exhibiting resistance to at least one drug in three or more antimicrobial categories [23], and 94.1% exhibited FQR.

PCR analysis was conducted to detect the presence of *bla*_ESBL_ (SHV, CTX-M), *bla*_CMY_, and *bla*_CARBA_ (VIM, IMP, OXA, NDM, KPC) genes, followed by Sanger sequencing. The CTX-M-15 gene was identified in 94.1% of the isolates, with one isolate harbouring the CTX-M-194 variant. Notably, the *bla*_NDM_ gene was identified in the two *K. pneumoniae* isolates that exhibited ertapenem resistance, which also carried the CTX-M-15 gene. Additionally, two isolates possessed both the CTX-M-15 and SHV-148 genes. None of the isolates tested positive for *mcr* (1 to 5) genes or showed polymyxin resistance (Appendix A).

#### 2.2.2. Virulence Traits

The hypermucoviscous (HMV) phenotyping of the 17 *K. pneumoniae* isolates, using the string test described by Shon et al. [35] with modifications, showed that four were classified as HMV-positive in at least two of the tested culture conditions (Table 4 and Appendix A). Notably, two isolates exhibited the HMV-positive phenotype across all culture conditions.

PCR screening was conducted to detect virulence genes commonly associated with hypervirulent *K. pneumoniae* (hvKp) [36]. All isolates tested negative for *iroB*, *peg-589*, *peg-1631*, and *rmpA2* genes. However, more than 50% were positive for *terB* and *rpmA*. Notably, the two *K. pneumoniae* isolates exhibiting the HMV phenotype across all culture conditions tested positive for *iucA*, *peg-344*, and *rmpA* genes, and co-harboured both *bla*_NDM_ and *bla*_CTX-M-15_ (Table 4).

## 3. Discussion

Urinary tract infections (UTIs) continue to represent a significant public health concern globally, with the pathogens *E. coli* and *K. pneumoniae* being the primary causative agents [1,4]. Particularly in Algeria, UTIs and pyelonephritis were reported as the fifth leading cause of death in 2021, accounting for 1,290 fatalities. *E. coli* and *K. pneumoniae* were the primary pathogens responsible for 40.5% of the total deaths (370 and 153, respectively). Among these, 84% of deaths were attributed to resistant *E. coli* and *K. pneumoniae* (321 and 117, respectively) [37]. Given the significant role of these bacteria in the aetiology of UTIs, their global contribution to the dissemination of MDR, and the limited data on their impact in Algeria, this study aimed to investigate the molecular characteristics of *E. coli* and *K. pneumoniae* exhibiting ESC resistance associated with UTIs in the province of Tebessa, where data on such infections remain scarce. Although our study was limited to three healthcare facilities in Tebessa, the selected sites are key referral centers that serve a diverse patient population across the province, making the sample reasonably representative of urban Algeria. The Laboratory of Medical Analysis receives a large and diverse influx of patients from across the province, due to its advanced equipment and strong reputation among local physicians who frequently refer their patients there. The Bouguerra Boulares Hospital and Khaldi Abdelaziz Maternity Hospital are central healthcare institutions in the province, receiving patients from various areas, particularly in complex or high-risk cases. Diagnostic approaches and antimicrobial prescribing practices in these facilities mirror those used in other regions of the country, where antibiotics remain widely accessible without prescription, contributing to similar AMR dynamics.

CTX-M-15 and OXA-48-like enzymes are the most prevalent and widely disseminated ESBLs and carbapenemases, respectively, posing a significant global public health threat [38,39]. In this study, *bla*_CTX-M-15_ was detected in 87.5% of *E. coli* isolates, a prevalence similar to that reported in other regions of Algeria for ESBL-producing *E. coli* in UTIs [19,20]. The *bla*_OXA-48_ carbapenemase gene, endemic in North Africa and highly reported in Algeria [21], was found here in three *E. coli* isolates.

Notably, we identified two *bla*_CTX-M-27_
*E. coli isolates*, which also harboured *bla*_OXA-48_; this would represent the first report of the CTX-M-27 variant in Algeria, whose emergence and global spread have been reported in regions such as Japan and Europe [38,40]. The third OXA-48 *E. coli* isolate analysed here showed a co-carriage of CMY-59. Furthermore, we observed FQR in 62.5% of ESC-producing *E. coli*, which aligns with findings from Sétif [19] but is higher than the 30% resistance prevalence reported by Zenati et al. [20] in Tlemcen, Algeria.

The *E. coli* population is classified into distinct phylogenetic groups, namely A, B1, B2, C, D, F, and G, along with cryptic clades [26,27]. UPEC isolates predominantly belong to phylogroups B2 and D, which are typically associated with the carriage of a higher number of virulence genes in comparison with other phylogroups [41,42]. Accordingly, our ESC-producing *E. coli* isolates were classified into A, B1, B2, D, and F, with B2 being the most prevalent (52.5%), notably including high-risk clones such as CC131 and ST1193. Phylogroup D was the second, comprising 22.5% of the isolates. Of the 21 B2 isolates, 19 met the criteria for UPEC status, and 20 for ExPEC status. Additionally, all nine phylogroup D isolates fulfilled the ExPEC status.

The CC131 and ST1193 are reported as the most prevalent among FQ and cephalosporin-resistant *E. coli* isolates globally, commonly linked to MDR UTIs [10,11,43,44]. The CC131 lineage of *E. coli* has diversified into three clades: A and B, which are susceptible, and C, which exhibits FQ/cephalosporin resistance. Clade C, particularly the *fimH*30 variant (CH40-30 clonotype), is the most widely distributed. Within clade C, subclade C1 (H30*R*) includes FQR non-ESBL producers, whereas subclade C2 (H30*Rx*) is characterized by the presence of both FQR and the *bla*_CTX-M-15_ gene [45].

In this study, we used *rfb*O25b, *fli*C_H4_, *fli*C_H5_, and the B2 phylogroup as PCR screening markers for CC131, along with clonotyping, as previously described [10]. This approach allowed us to identify 17 O25b:H4-B2 *E. coli* (42.5%), all assigned to CC131 and exhibiting the CH40-30 clonotype. Moreover, all isolates displayed FQR and carried the *bla*_CTX-M-15_ gene, classifying them within subclade C2 [45].

CC131 has been reported in various sources in Algeria, including human clinical samples [46,47], uropathogenic *E. coli* from non-hospitalized patients [48], fish [49], food [50], and wildlife [51], indicating its widespread distribution across the country. However, detailed molecular characterization of CC131 in Algeria remains limited. Our study focuses on resistance and virulence gene profiles. Thus, we found that 88.2% and 94.1% of CC131 isolates exhibited UPEC and ExPEC status, respectively. Furthermore, analysis of extraintestinal virulence factors following the Dahbi et al. [30] scheme identified two distinct virotypes: F (*sat*, *papG II*, *kpsM II-K5*) in 41.2% of isolates and E (*sat*, *papG II*, *cnf1*, *hlyA*, *kpsM II-K5*) in 17.6%. Notably, the remaining seven FQR, CTX-M-15 O25b:H4-B2-CC131 (CH40-30) isolates could not be virotyped.

The ST1193 clone is considered an emerging pathogenic lineage of *E. coli* [52,53] typically associated with the O75 serotype, *fimH*64 type 1 pili, and K1 or K5 capsular types. This clone is also characterized by FQR and lactose non-fermentation [11]. Although ST1193 has been reported in Europe [9,10,54,55], Asia [56,57,58], and the United States [59], its detection in Africa remains limited [60,61,62]. In this study, we identified one NLF ST1193 isolate, FQR, CTX-M-15-producer, with genomic key features consistent with globally disseminated ST1193 isolates [9,10,11]. Specifically, the Algerian isolate carried the K1 capsular type and harboured mutations in *gyrA* (D87N, S83L) and *parC* (S80I), along with *parE* (L416F). Plasmidome analysis identified the presence of IncF [F-:A1:B10], IncI1-I, and ColBS512-like plasmids. A large-scale genomic study of ST1193 [52] classified the ST1193 clone into distinct clades based on capsule type (K1 or K5) and the presence of specific IncF plamids: clade A (K5 capsule with F−:A1:B20 plasmids), subclade B1 (K1 capsule with F−:A1:B10 plasmids) and subclade B2 (K1 capsule with F−:A1:B1 plasmids) [11,52]. According to this classification, our ST1193 isolate belongs to subclade B2.

A core genome SNP comparison between our Algerian ST1193 isolate (LREC-468) and previously characterized ST1193 isolates from a Northwest Spanish hospital [9] revealed a minimum distance of 78 SNPs with LREC-269. We then queried Enterobase for genomes assigned to cgST140226 and found a single genome, ESC_RA5887AA, from BioProject PRJEB21277 (University of Oxford, 2020), which differed by only 55 SNPs.

This comparative approach was guided by the core genome sequence typing (cgST) framework proposed by Achtman et al. [63], which offers a robust method for identifying highly related strains based on shared allelic profiles. While hundreds of ST1193 genomes are available in public repositories, few are annotated with cgST identifiers, and fewer still meet the quality criteria and metadata completeness required for SNP-level comparisons. cgMLST, which analyzes 2,513 soft-core genes, offers significantly higher resolution than MLST, making it a powerful tool for tracing transmission dynamics within outbreaks and defining population structures across different levels, including the genus level [63]. Interestingly, the detection of the same cgST in a genome presumably originating from Ireland (based on the metadata provided in the BioProject of this genome) underscores the transcontinental dissemination of this clone and raises public health concerns about the unnoticed spread of high-risk *E. coli* lineages in North Africa. Our findings, therefore, emphasize the urgent need for continued genomic surveillance of MDR uropathogens in the region, particularly as this is the first genomic report of ST1193 in clinical *E. coli* isolates from Algeria.

Phylogroup D was the second most prevalent in our *E. coli* collection, comprising 22.5% of the isolates, all classified as ExPEC. These isolates exhibited either the *fum*C26 *fim*H5 or *fum*C26 *fim*H-negative clonotypes (no amplification of the 489-bp internal sequence of the clonotype scheme) [31]. According to EnteroBase, all genomes assigned to phylogroup D (Clermont scheme) with *fum*C26 *fim*H5/negative are associated with CC38, which includes ST38, a high-risk MDR clone linked to UTI and bloodstream infections, as well as the global spread of OXA-48 [39]. ST38 has also contributed to the emergence of nosocomial and community-acquired OXA-244-producing *E. coli* ST38 in Europe [64,65]. Additionally, putative inter-host and host-environment transmission events within ST38, where genomes differed by <35 SNPs, underscore its role in maintaining and disseminating AMR genes [66]. In this study, the two D-CH26-Neg isolates carried both OXA-48-like and CTX-M-27, while the one D-CH26-5 isolate harboured OXA-48-like and CMY-59. ST38 OXA-48-producing *E. coli* has previously been detected in Algeria in white stork (*Ciconia ciconia*) migratory birds [67], river water [68], human clinical samples [69], and broilers [70]. Our findings further highlight the role of CC38 in the spread of AMR in this region.

All *K. pneumoniae* isolates, except one, carried the *bla*_CTX-M-15_ gene. In addition, two ertapenem-resistant isolates tested positive for the *bla*_NDM_ gene. The co-occurrence of carbapenem and ESBL resistance in the same isolate significantly complicates treatment options and raises concerns about therapeutic failures. Specifically, *K. pneumoniae* harbouring *bla*_CTX-M-15_ and *bla*_NDM_ genes was previously reported at Annaba University Hospital (Algeria) in 2014 [71]. Further studies have reported the widespread presence of *bla*_CTX-M-15_ in Algeria, including the University Hospital Establishment of Oran, where nearly all ESBL-positive isolates carried *bla*_CTX-M-15_ [72]. *bla*_CTX-M_ *K. pneumoniae* isolates were also identified in the Regional Military University Hospital of Oran, with a prevalence of 37.5% ESBL producers [17]. Carbapenem-resistant *K. pneumoniae* strains carrying *bla*_CTX-M-15_ have also been documented in Ibn Roched hospital of Annaba, where OXA-48 and KPC-2 carbapenemase-producing isolates were identified in urology patients [16]. These findings underscore the ongoing emergence of multidrug-resistant *K. pneumoniae* in Algerian hospitals, necessitating enhanced surveillance and infection control measures.

In terms of virulence, *K. pneumoniae* is categorized into two distinct pathotypes: classical (cKp) and hypervirulent (hvKp). cKp is a common cause of hospital-acquired infections such as pneumonia and UTIs, particularly in elderly or immunocompromised people, and is known for its ability to acquire multiple AMR genes [73]. By contrast, hvKp is more virulent and capable of causing severe infections in healthy people, such as pyogenic liver abscesses, endophthalmitis, meningitis, septic arthritis, and other unusual infections, often leading to significant morbidity and mortality [74,75]. The definition of hvKp has evolved. Initially, the HMV phenotype with a positive string test (>5 mm) was used for identification [35,76], but this method lacks accuracy. Currently, murine infection models remain the gold standard [77]; however, their high cost and ethical constraints limit their practicality. Instead, recent evidence supports the presence of five virulence plasmid-associated genes (*peg-344*, *rmpA*, *rmpA2*, *iroB*, and *iucA*) as the most accurate molecular markers for hvKp identification, which offer a feasible diagnostic alternative for clinical and surveillance applications [78,79]. In our *K. pneumoniae* collection, the HMV phenotype was observed in four isolates, though only two were consistently positive across all test conditions. PCR screening of the virulence markers revealed that these two isolates were the only ones carrying three of the five plasmid-associated genes (*peg-344*, *rmpA*, and *iucA*). Notably, both also harboured the *bla*_NDM_ and *bla*_CTX-M-15_ genes, further highlighting their clinical relevance.

Initially, hvKp isolates were susceptible to common antibiotics, including last generation cephalosporins and carbapenems. However, the emergence of MDR hvKp in recent years has raised significant concerns. Notably, the European Centre for Disease Prevention and Control reported the spread of hvKp ST23 strains carrying carbapenemase genes across multiple EU/EEA countries, highlighting the convergence of hypervirulence and antimicrobial resistance [80]. Although our two HMV *bla*_NDM_ and *bla*_CTX-M-15_-producing *K. pneumoniae* possessed only three of the five key virulence plasmid-associated genes (*peg-344*, *rmpA*, and *iucA*), they exemplify the concerning trend of MDR *K. pneumoniae* strains enhancing their clinical virulence potential. This convergence of resistance and virulence factors poses significant challenges for treatment and infection control strategies.

In Algeria, the widespread and often unregulated use of antibiotics, especially beta-lactams, continues to drive the emergence and spread of ESBL-producing bacteria. In Tebessa province, a recent study confirmed the strong correlation between antibiotic misuse and the growing prevalence of resistant pathogens [18]. Similar findings from northeastern [19] and northwestern Algeria [20] linked high resistance rates to inappropriate prescribing practices and over-the-counter antibiotic sales without medical oversight. These trends highlight the urgent need for robust antibiotic stewardship and strict regulation of antimicrobial use nationwide.

AMR data in Algeria remains limited and fragmented. As noted by Touati and Mairi [21], most available evidence stems from isolated academic efforts, often in collaboration with European labs, rather than from coordinated national surveillance. The Algerian Antimicrobial Resistance Network (AARN) [81] has also reported increasingly alarming resistance patterns in *E. coli* and *K. pneumoniae* from hospitals across the country, emphasizing challenges such as diagnostic variability and under-resourced microbiology services.

In this context, our study contributes valuable region-specific data by characterizing the virulence and antibiotic resistance profiles of *E. coli* and *K. pneumoniae* strains causing UTIs in Tebessa. By exploring a simplified laboratory workflow for AMR monitoring, this work supports the development of cost-effective surveillance tools. A clearer understanding of local epidemiological patterns is essential to guide empirical therapy, inform infection control strategies, and strengthen Algeria’s national AMR response.

## 4. Materials and Methods

### 4.1. E. coli and K. pneumoniae Collections

This study analyzed 57 non-duplicate isolates, comprising 40 *E. coli* and 17 *K. pneumoniae*, recovered from urine samples of both male (9 *E. coli* and 5 *K. pneumoniae*) and female (31 *E. coli* and 12 *K. pneumoniae*) patients. The samples were obtained from both outpatients and inpatients at three healthcare facilities in the Tebessa region of northeast Algeria. These facilities were the Bouguerra Boularess-Bekaria Public Hospital Establishment (EPH) (252 beds, 8 wards), the Khaldi Abdelaziz-Tebessa Maternity Hospital (EPH) (166 beds, 4 wards), and the private laboratory Elite. Isolates were collected between July 2022 and January 2024.

Following incubation on MacConkey (ML) agar (Oxoid, Sin El Fil, Beirut, Lebanon) at 37 °C for 18–24 h, bacterial identification was performed using the API 20E system (bioMérieux, Dely Ibrahim, Alger, Algeria) and the Vitek 2 GN system (bioMérieux Inc., Hazelwood, MO, USA). Isolates were selected based on extended-spectrum cephalosporin (ESC) resistance using the Vitek 2 AST system (bioMérieux, Dely Ibrahim, Alger, Algeria). Selected isolates were stored on slanted nutrient agar (Difco, Sin El Fil, Beirut, Lebanon) at room temperature until further molecular analysis at the Reference Laboratory for *E. coli* (LREC, University of Santiago de Compostela, Spain) (Appendix A).

Species confirmation was conducted by PCR amplification of the β-d-glucuronidase (*uidA*) gene for *E. coli* [22] and the putative acyltransferase (*kp50233*) gene for *K. pneumoniae* [82] (Appendix A). When PCR results were inconclusive, bacterial identification was performed using matrix-assisted laser desorption/ionization–time-of-flight mass spectrometry (MALDI-TOF) (Bruker Daltonik, Bremen, Germany). A species-level identification was considered reliable only if the obtained score exceeded 2.

Non-lactose fermenting (NLF) *E. coli* were identified phenotypically based on their inability to ferment lactose after overnight incubation on ML agar at 37 °C.

### 4.2. Antimicrobial Susceptibility Testing (AST)

At the LREC (University of Santiago de Compostela, Spain), antimicrobial susceptibility testing (AST) was performed using the disc diffusion method (Becton Dickinson, Sparks, MD, USA) on Mueller–Hinton (MH) agar (Oxoid, Madrid, Spain). A total of 20 antibiotics were tested: penicillin (ampicillin), penicillin + beta-lactamase inhibitors (amoxicillin-clavulanic acid), non-broad spectrum cephalosporins (cefuroxime); broad-spectrum cephalosporins (cefoxitin, cefotaxime, ceftazidime), carbapenems (ertapenem), monobactams (aztreonam), fluoroquinolones (nalidixic acid, ciprofloxacin), aminoglycosides (amikacin, gentamicin, tobramycin), tetracyclines (doxycycline), glycylcyclines (tigecycline), phosphonic acids (fosfomycin), nitrofurans (nitrofurantoin), folate pathway inhibitors (trimethoprim-sulfamethoxazole), polymyxins (colistin), and amphenicols (chloramphenicol).

AST results were interpreted according to the European Committee on Antimicrobial Susceptibility Testing [83] clinical breakpoints when available, or Clinical & Laboratory Standards Institute [84] as an alternative. Isolates were classified as MDR if they exhibited resistance to at least one drug in three or more antimicrobial categories [23].

### 4.3. Detection and Typing of Antimicrobial Resistance Genes

The isolates were screened by PCR for clinically relevant *bla*_ESC_ genes using primers specific for SHV, CMY, CTX-M, VIM, IMP, OXA-48, KPC, and NDM. Sanger sequencing was performed when appropriate, in order to confirm gene variants or subtyping, as described elsewhere [85,86,87,88,89,90,91] (Appendix A). In addition, mobile colistin resistance genes *mcr-1* to *mcr-5* were screened by PCR [92,93] (Appendix A).

### 4.4. Molecular Characterization of E. coli: Virulence Traits, Phylogroup, Clonotype, and Virotype Assignment

For the molecular characterization of the *E. coli* collection, we followed the workflow scheme proposed by García-Meniño et al. [10]. Briefly, specific virulence-associated genes statistically linked to increased efficiency in urinary tract colonization were tested by PCR. Isolates were classified as UPEC if they harbored ≥3 of the following genes: *chuA*, *fyuA*, *vat*, or *yfcV* [24] (Appendix A). The designation of extraintestinal pathogenic *E. coli* (ExPEC) status was attributed to isolates positive for ≥2 of these five markers: *papAH*, *sfa/focDE*, *afa/draBC*, *kpsM II,* or *iutA* [25,33,94] (Appendix A).

The clonal structure of the *E. coli* collection was investigated using the phylogroup assignment method of Clermont et al. [26,27,95] (Appendix A), which differentiates eight *E. coli* phylogroups (A, B1, B2, C, D, E, F, and G). Clonotype (CH) determination was performed by Sanger sequencing a 469-nucleotide (nt) internal region of the *fumC* gene (allele derived from MLST) and a 489-nt internal fragment of the *fimH* gene, encoding type 1 fimbrial adhesion [31] (Appendix A).

To presumptively identify the pandemic CC131 lineage, we screened for phylogroup B2 along with *rfb*O25, *fliC*_H4_, and *fliC*_H5_. In addition, the *fliC*_H5_ flagellar-encoding gene is commonly associated with NFL *E. coli* ST1193 (Appendix A) [96]. Isolates confirmed as O25b:H4-B2-CH40-30 were further characterized for their virotypes according to the protocol established by Dahbi et al. [30], based on the presence or absence of specific extraintestinal virulence factors (*afa*/*draBC*, *afa* operon FM955459, *iroN*, *sat*, *ibeA*, *papG II*, *papG III*, *cnf1*, *hlyA*, *cdtB*, *kpsM II*-K1, -K2, and -K5) (Appendix A) [97,98,99,100,101,102].

### 4.5. Phenotypic and Genotypic Detection of Hypervirulent K. pneumoniae

Hypervirulent *K. pneumoniae* (hvKp) isolates were initially screened based on the hypermucoviscous (HMV) phenotype using the string test, as described by Shon et al. [35]. String tests were performed on colonies grown on MH agar, ML agar, trypticase soy (TSA) agar (Oxoid, Madrid, Spain), and Columbia agar (5% sheep blood) (Oxoid, Madrid, Spain) incubated at 37 °C/24 h. A positive string test was defined as the formation of a viscous string ≥ 5 mm in length when a bacteriological inoculation loop was used to stretch bacterial colonies on an agar plate [35].

Additionally, the presence of molecular markers statistically associated with hvKp was assessed by PCR. Isolates were classified as hvKp if they carried all five hvKp virulence plasmid-associated genes: *peg-344* (putative transporter), *rmpA* and *rmpA2*(regulators of the mucoid phenotype via increased capsule production), *iroB* (salmochelin siderophore biosynthesis), and *iucA* (aerobactin siderophore biosynthesis) [36,78] (Appendix A).

### 4.6. Whole Genome Sequencing (WGS) and Bioinformatics Analysis

The *E. coli* isolate classified as *fliC*_H5_-B2 and CH14-64 was further analysed by WGS as described by García et al. [103]. Briefly, DNA was extracted with the DNeasey Blood and Tissue Kit (Qiagen, Hilen, Germany) according to the manufacturer’s instructions. After extraction, the DNA was quantified by an Invitrogen Qubit fluorimeter (Thermo Fisher Scientific, Waltham, MA, USA) and evaluated for purity using a NanoDrop ND-1000 (Thermo Fisher Scientific, Waltham, MA, USA). The DNA sequencing was performed using Illumina technology with a NovaSeq 6000 S4PE150 XP system to obtain 150 bp paired-end reads at Eurofins Genomics (Eurofins Genomics GmbH, Konstanz, Germany), following library preparation using the standard Illumina DNA Prep kit (Illumina Cambridge Ltd., Cambridge, UK). The quality of the paired-end Illumina reads was evaluated using FastQC v0.12.1.

The genome reconstruction and in silico analysis were performed as described elsewhere [104]. Briefly, the raw reads were assembled with the VelvetOptimiser.pl. script implemented in the online version of the PLAsmid Constellation NETwork (PLACNETw). The assembled contigs, with genomic size 5.0 Mbp, were analyzed using the bioinformatics tools of the Center for Genomic Epidemiology (CGE) as specified, and applying the thresholds suggested by default when required (minimum identity of 95% and coverage of 60%) for the presence of acquired genes and or chromosomal mutations mediating antimicrobial resistance (ResFinder 4.6.) [105,106]; for identification of acquired virulence genes (VirulenceFinder 2.0) [107,108]; plasmid replicon types (PlasmidFinder 2.1/pMLST 2.0) [109]; identification of clonotypes (CHTyper 1.0) [110]; and serotypes (SeroTypeFinder 2.0.1) [111]. Two different MLST (2.0.9) schemes were applied for phylogenetic typing [34,112]. Additionally, cgMLSTFinder1.2 was called for the core genome multi-locus typing (cgMLST) [106,113].

To investigate the phylogenetic relationship of the ST1193 isolate sequenced in this study, a comparative analysis was performed using CSI Phylogeny 1.4 with five previously described ST1193 isolates from a hospital in northwest Spain [9].

The pipeline was run, using the genome of LREC-269 as the reference for single-nucleotide polymorphism (SNP) calling, with the following parameters: min. depth at SNP positions ×10; min. relative depth at SNP positions: ×10; min. distance between SNPs (prune): 10 bp; min. SNP quality: 30; min. read mapping quality: 25, a min. Z-score of 1.96 and by ignoring heterozygous SNPs). Branch values represent substitutions per site. Bootstrap support for the consensus phylogenetic tree was calculated using 1000 replicates [114]. The resulting SNP matrix is shown in Appendix A.

Finally, EnteroBase (https://EnteroBase.warwick.ac.uk/; accessed on 20 December 2024) was queried for ST1193 genomes based on the Achtman 7-gene MLST scheme. Specific core genome MLST (cgMLST) sequences were retrieved, and their raw reads were used for comparative genomic analysis (Appendix A).

## 5. Conclusions

This study provides important insights into the epidemiology of MDR *E. coli* and *K. pneumoniae* in Algeria, highlighting the presence and diversity of *E. coli* CC131 CH40-30 and reporting, for the first time, *E. coli* ST1193 carrying CTX-M-15 in this region. The detection of virulent *K. pneumoniae* co-harbouring carbapenemase and ESBL resistance genes is particularly concerning, as it raises the potential for increased pathogenicity and treatment failure in clinical settings.

The co-occurrence of AMR and virulence factors underscores the need for enhanced surveillance and infection control measures. A simplified surveillance method based on virulence traits in *E. coli* and *K. pneumoniae* is proposed for early detection and outbreak monitoring. In addition, monitoring high-risk clones such as CC38 and OXA-48 is crucial for preventing further public health threats. A practical lab workflow for identifying high-risk *E. coli* clones associated with UTIs is outlined, emphasizing non-lactose fermenting isolates, antimicrobial susceptibility testing, and phylogroup/clonotyping.

This study had certain limitations, as the isolates were collected from only three healthcare facilities within a single province (Tebessa), which may limit the broader geographical representation of the findings. The sample size, while sufficient for exploratory analysis, may not fully capture the genetic and phenotypic diversity of *E. coli* and *K. pneumoniae* circulating in other regions of Algeria. Nevertheless, our findings are consistent with previous reports from northern and western Algeria, suggesting that the AMR patterns observed in uropathogens may reflect broader national trends. Furthermore, future studies involving in-depth genomic characterization, such as plasmid profiling and mobility analyses, are needed to understand the dynamics of resistance gene dissemination.

Given the significant resistance profiles observed here, targeted antibiotic optimization programs should focus on rational antibiotic use, improved prescription practices, and enhanced surveillance, especially in LMICs where MDR infections pose a major public health burden.

## Figures and Tables

**Figure 1 antibiotics-14-00485-f001:**
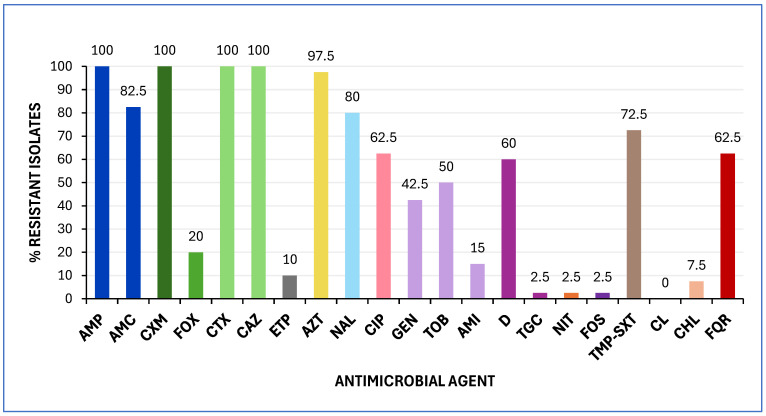
Prevalence of antimicrobial resistance among the 40 *E. coli* isolates analysed in this study. Minimum inhibitory concentrations (MICs) were interpreted according to EUCAST 2025 and CLSI 2024 clinical breakpoints. Abbreviations: AMP, ampicillin; AMC, amoxicillin/clavulanic acid; CXM, cefuroxime; FOX, cefoxitin; CTX, cefotaxime; CAZ, ceftazidime; ETP, ertapenem; AZT, aztreonam; NAL, nalidixic acid; CIP, ciprofloxacin; GEN, gentamicin; TOB, tobramycin; AMI, amikacin; D, doxycycline; TGC, tigecycline; NIT, nitrofurantoin; FOS, fosfomycin; TMP-SXT, trimethoprim-sulfamethoxazole; CL, colistin; CHL, chloramphenicol; FQR, fluoroquinolone resistance.

**Figure 2 antibiotics-14-00485-f002:**
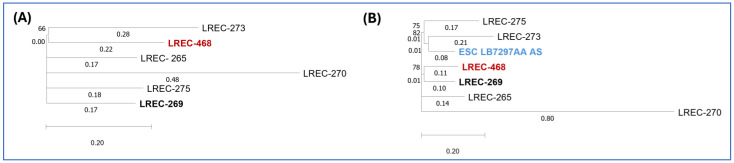
Phylogenetic dendrogram based on whole-genome SNP analysis. (**A**) Phylogenetic dendrogram based on SNP counts per substitution within the core genome of the Algerian and Spanish ST1193 isolates. The WGS comparison resulted in a core genome covering 95.65% of the reference genome LREC-269 (5.4 Mb). (**B**) Phylogenetic dendrogram incorporating the Algerian, Spanish, and ESC_RA5887AA ST1193 isolates, showing SNP counts per substitution. The WGS comparison resulted in a core genome covering 82.6% of the reference genome LREC-269 (5.4 Mb).

**Figure 3 antibiotics-14-00485-f003:**
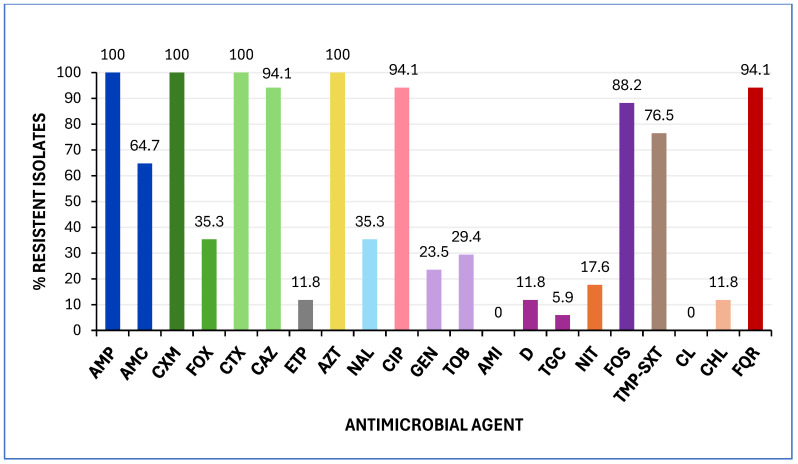
Prevalence of antimicrobial resistance among the 17 *K. pneumoniae* analysed in this study. MICs were interpreted according to EUCAST 2025 and CLSI 2024 clinical breakpoints. Abbreviations: AMP, ampicillin; AMC, amoxicillin/clavulanic acid; CXM, cefuroxime; FOX, cefoxitin; CTX, cefotaxime; CAZ, ceftazidime; ETP, ertapenem; AZT, aztreonam; NAL, nalidixic acid; CIP, ciprofloxacin; GEN, gentamicin; TOB, tobramycin; AMI, amikacin; D, doxycycline; TGC, tigecycline; NIT, nitrofurantoin; FOS, fosfomycin; TMP-SXT, trimethoprim-sulfamethoxazole; CL, colistin; CHL, chloramphenicol; FQR, fluoroquinolone resistance.

**Table 2 antibiotics-14-00485-t002:** Phenotypic resistance and virulence profile of the 17 CC131 *E. coli* isolates.

Isolate	UPEC Status ^a^	ExPEC Status ^b^	Virotype ^c^	Phenotypic Resistance ^d^	Virulence Genes Profile	*bla* Genes
Ec4	+	+	NT	AMP, CXM, CTX, CAZ, AZT, NAL, CIP, TOB, AMI, TMP-SXT	*fyuA*, *yfcV*, *chuA*, *papAH*, *iutA*, *kpsMII-*K5, *papGII*	CTX-M-15
Ec5	+	+	NT	AMP, AMC, CXM, CTX, CAZ, AZT, NAL, CIP, TOB, AMI, D, TMP-SXT	*fyuA*, *yfcV*, *chuA*, *papAH*, *iutA*, *kpsMII-*K5, *papGII*	CTX-M-15
Ec8	+	+	F	AMP, AMC, CXM, CTX, CAZ, AZT, NAL, CIP, GEN, TOB, AMI, TMP-SXT	*fyuA*, *yfcV*, *chuA*, *papAH*, *iutA*, *kpsMII-*K5, *sat*, *papGII*	CTX-M-15
Ec10	+	+	F	AMP, AMC, CXM, CTX, CAZ, AZT, NAL, CIP, GEN, TOB	*fyuA*, *yfcV*, *chuA*, *papAH*, *iutA*, *kpsMII-*K5, *sat*, *papGII*	CTX-M-15
Ec19 a	+	+	F	AMP, AMC, CXM, CTX, CAZ, AZT, NAL, CIP, GEN, TOB	*fyuA*, *yfcV*, *chuA*, *papAH*, *iutA*, *kpsMII-*K5, *sat*, *papGII*	CTX-M-15
Ec20 a	+	+	E	AMP, AMC, CXM, CTX, CAZ, AZT, NAL, CIP, GEN, TOB	*fyuA*, *yfcV*, *chuA*, *papAH*, *iutA*, *kpsMII-*K5, *sat*, *papGII*, *cnfI*, *hlyA*	CTX-M-15
Ec22	+	+	F	AMP, CXM, CTX, CAZ, AZT, NAL, CIP, GEN, TOB, TMP-SXT	*fyuA*, *yfcV*, *chuA*, *papAH*, *iutA*, *kpsMII-*K5, *sat*, *papGII*	CTX-M-15
Ec23	+	−	NT	AMP, AMC, CXM, CTX, CAZ, AZT, NAL, CIP, TOB, AMI, D, NIT, TMP-SXT	*fyuA*, *yfcV*, *chuA*, *papAH*, *papGII*	CTX-M-15
Ec28	+	+	NT	AMP, AMC, CXM, CTX, CAZ, AZT, NAL, CIP, GEN, TOB, TMP-SXT	*fyuA*, *yfcV*, *chuA*, *papAH*, *iutA*, *sat*, *papGII*	CTX-M-15
Ec30	+	+	NT	AMP, AMC, CXM, CTX, FOX, CAZ, ETP, AZT, NAL, CIP, TMP-SXT	*fyuA*, *yfcV*, *chuA*, *papAH*, *iutA*, *sat*, *papGII*	CTX-M-15
Ec31	+	+	NT	AMP, AMC, CXM, CTX, FOX, CAZ, AZT, NAL, CIP, GEN, TOB, D, TMP-SXT	*fyuA*, *yfcV*, *chuA*, *papAH*, *iutA*, *sat*, *papGII*	CTX-M-15
Ec41	−	+	F	AMP, AMC, CXM, CTX, CAZ, AZT, NAL, CIP, GEN, TOB, D, TMP-SXT	*chuA*, *iutA*, *kpsMII-*K5, *sat*, *papGII*	CTX-M-15
Ec43	+	+	F	AMP, AMC, CXM, CTX, CAZ, AZT, NAL, CIP, GEN, TOB, D TMP-SXT	*fyuA*, *yfcV*, *chuA*, *papAH*, *iutA*, *kpsMII-*K5, *sat*, *papGII*	CTX-M-15
Ec45	+	+	NT	AMP, AMC, CXM, CTX, CAZ, AZT, NAL, CIP, GEN, TOB, AMI, D, TMP-SXT	*fyuA*, *yfcV*, *chuA*, *papAH*, *iutA*, *sat*, *papGII*	CTX-M-15
Ec52	+	+	F	AMP, AMC, CXM, CTX, CAZ, AZT, NAL, CIP, GEN, TOB, D TMP-SXT	*fyuA*, *yfcV*, *chuA*, *papAH*, *iutA*, *kpsMII-*K5, *sat*, *papGII*	CTX-M-15
Ec55 a	+	+	E	AMP, AMC, CXM, CTX, CAZ, AZT, NAL, CIP, GEN, TOB	*fyuA*, *yfcV*, *chuA*, *papAH*, *iutA*, *kpsMII-*K5, *sat*, *papGII*, *cnfI*, *hlyA*	CTX-M-15
Ec57	−	+	E	AMP, AMC, CXM, CTX, FOX, CAZ, AZT, NAL, CIP, GEN, TOB, D, TGC, FOS, TMP-SXT, CHL	*yfcV*, *chuA*, *papAH*, *iutA*, *kpsMII-*K5, *sat*, *papGII*, *cnfI*, *hlyA*	CTX-M-15

^a^ UPEC status. + positive for ≥3 of the following genes: *chuA*, *fyuA*, *vat*, and *yfcV* [24]; otherwise, negative. ^b^ ExPEC status. + positive for ≥2 of these five markers: *papAH*, *sfa/focDE*, *afa/draBC*, *kpsMII,* and *iutA* [25]; otherwise, negative. ^c^ Virotypes following the protocol established by Dahbi et al. [30], based on the presence or absence of specific extraintestinal virulence factors (*afa*/*draBC*, *afa* operon FM955459, *iroN*, *sat*, *ibeA*, *papGII*, *papGIII*, *cnf1*, *hlyA*, *cdtB*, *kpsMII*-K1, -K2 and -K5); NT, not typable. ^d^ Abbreviations: AMP, ampicillin; AMC, amoxicillin/clavulanic acid; CXM, cefuroxime; FOX, cefoxitin; CTX, cefotaxime; CAZ, ceftazidime; ETP, ertapenem; AZT, aztreonam; NAL, nalidixic acid; CIP, ciprofloxacin; GEN, gentamicin; TOB, tobramycin; AMI, amikacin; D, doxycycline; TGC, tigecycline; NIT, nitrofurantoin; FOS, fosfomycin; TMP-SXT, trimethoprim-sulfamethoxazole; CL, colistin; CHL, chloramphenicol.

**Table 3 antibiotics-14-00485-t003:** In silico characterization and phenotypic resistance of the ST1193 genome.

ID Code for Isolate/Genome ^a^	Ec1a/LREC-468
**O:H antigens ^b^**	O75:H5
**ST#1/ST#2 ^c^**	1193/53
**cgMLST ^d^**	140226
**Acquired resistance and** **point mutations [in brackets] ^e^**	*aac(3)-IIa*, *aac(6′)-Ib-cr*, *bla*_CTX-M-15_, *bla*_OXA-1_, *catB3* [*gyrA* p.S83L, *gyrA* p.D87N, *parC* p.S80I, *parE* p.L416F]
**Plasmid content: Inc. group [pMLST] ^f^**	IncF [F-:A1:B10], IncI1-I [ST Unknown], ColBS512-like
**Virulence genes ^g^**	*aslA*, *chuA*, *cia*, *csgA*, *fdeC*, *fimH*, *fyuA*, *gad*, *iha*, *irp2*, *iucC*, *iutA*, *kpsE*, *kpsMII_K1*, *neuC*, *nlpI*, *ompT*, *papA_F43*, *sat*, *shiA*, *sitA*, *terC*, *tia*, *usp*, *vat*, *yehA*, *yehB*, *yehC*, *yehD*, *yfcV*
**Phenotypic resistance ^h^**	AMP, AMC, CXM, CAZ, CTX, ATM, NAL, CIP, GEN

^a^ Isolate and genome (LREC) identification. ^b^ O and H antigen prediction with SerotypeFinder 2.0.1. ^c^ Sequence types (ST#1 and ST#2) based on two different MLST schemes *E. coli* #1 and *E. coli* #2, respectively, and retrieved with MLST 2.0.9. ^d^ Core genome ST obtained with cgMLSTFinder 1.2 run against the Enterobase database. ^e^ Acquired antimicrobial resistance genes retrieved using ResFinder 4.6: Acquired resistance genes: beta-lactam: *bla*_OXA-1_, *bla*_CTX-M-15_; aminoglycosides: *aac(3)-IIa*; phenicols: *catB3*; fluoroquinolones: *aac(6′)-Ib-cr*. Point mutations: quinolones and fluoroquinolones: *gyrA* S83L: TCG-TTG, *gyrA* D87N: GAC-AAC, *parC* S80I: AGC-ATC, *parE* L416F: CTT-TTT. ^f^ Plasmid content retrieved using PlasmidFinder 2.1. ^g^ Virulence determinants via VirulenceFinder 2.0: *aslA*: putatitive sulfatase; *chuA*: outer membrane hemin receptor; *cia*: colicin ia; *csgA*: curlin major subunit CsgA; *fdeC:* intimin-like adhesin FdeC; *fimH:* type 1 fimbriae; *fyuA:* siderophore receptor; *gad*: glutamate decarboxylase; *iha*: adherence protein; *irp2*: high molecular weight protein 2 non-ribosomal peptide synthetase; *iucC*: aerobactin synthetase; *iutA*: ferric aerobactin receptor; *kpsE*: capsule polysaccharide export inner-membrane protein; *kpsMII_K1*: polysialic acid transport protein group 2 capsule; *neuC*: polysialic acid capsule biosynthesis protein; *nlpI*: lipoprotein NlpI precursor; *ompT*: outer membrane protease (protein protease 7); *papA_F43*: major pilin subunit F43; *sat*: secreted autotransporter toxin; *shiA*: homologs of the Shigella flexneri SHI-2 pathogenicity island gene shiA; *sitA*: iron transport protein; *terC*: tellurium ion resistance protein; *usp*: uropathogenic specific protein; *vat*: vacuolating autotransporter toxin; *yehA*: outer membrane lipoprotein, YHD fimbriael cluster; *yehB*: usher, YHD fimbriael cluster; *yehC*: chaperone, YHD fimbriael cluster; *yehD*: major pilin subunit, YHD fimbriael cluster; *yfcV*: fimbrial protein. ^h^ Abbreviations: AMP, ampicillin; AMC, amoxicillin/clavulanic acid; CXM, cefuroxime; CTX, cefotaxime; CAZ, ceftazidime; AZT, aztreonam; NAL, nalidixic acid; CIP, ciprofloxacin; GEN, gentamicin.

**Table 4 antibiotics-14-00485-t004:** Hypervirulent and AMR traits of the four phenotypic HMV *K. pneumoniae* isolates.

	Hypermucoviscous Phenotype (HMV) ^a^						
Isolate	ML	MH	TSA	CA	Phenotypic Resistance ^b^	Virulence Genes ^c^	*bla* Genes
KP6b	+	–	+	–	AMP, CXM, CTX, CAZ, AZT, CIP, FOS	*terB*	CTX-M-15
KP10c	+	+	+	+	AMP, AMC, CXM, FOX, CTX, CAZ, ETP, AZT, NAL, CIP, GEN, TOB, NIT, FOS	*iucA*, *peg-344*, *rmpA*	NDM, CTX-M-15
KP16	+	+	+	+	AMP, AMC, CXM, FOX, CTX, CAZ, ETP, AZT, NAL, CIP, GEN, TOB, TGC, NIT, FOS, TMP-SXT, CHL	*iucA*, *peg-344*, *rmpA*	NDM, CTX-M-15
KP20a	+	+	+	–	AMP, CXM, CTX, CAZ, AZT, CIP, FOS, TMP-SXT	*terB*	CTX-M-15

^a^ The HMV phenotype was evaluated using Mueller–Hinton (MH) agar, MacConkey (ML) agar, trypticase soy agar (TSA), and Columbia agar (5% sheep blood) (CA) medium culture. + and − indicates positive or negative result, respectively. ^b^ AMP, ampicillin; AMC, amoxicillin/clavulanic acid; CXM, cefuroxime; FOX, cefoxitin; CTX, cefotaxime; CAZ, ceftazidime; ETP, ertapenem; AZT, aztreonam; NAL, nalidixic acid; CIP, ciprofloxacin; GEN, gentamicin; TOB, tobramycin; TGC, tigecycline; NIT, nitrofurantoin; FOS, fosfomycin; TMP-SXT, trimethoprim-sulfamethoxazole; CHL, chloramphenicol. ^c^
*iucA*: aerobactin siderophore biosynthesis, *rmpA*: regulator of the mucoid phenotype via increased capsule production, *peg-344*: putative transporter, *terB*: tellurite resistance.

## Data Availability

The nucleotide sequence of the ST1193 *E. coli* isolate was deposited in the European Nucleotide Archive (ENA) with the following accession ERS22141751 and BioSample SAMEA117078805 codes, as part of BioProject ID PRJEB82513.

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
