# Peer review of "Phenotypic and Genotypic Characterization of ESBL-, AmpC-, and Carbapenemase-Producing *Klebsiella pneumoniae* and High-Risk *Escherichia coli* CC131, with the First Report of ST1193 as a Causative Agent of Urinary Tract Infections in Human Patients in Algeria"

_antibiotics, 2025, doi:10.3390/antibiotics14050485_

Round 1
Reviewer 1 Report
Comments and Suggestions for Authors
This manuscript presents a genomic epidemiology investigation of bacterial isolates collected from healthcare facilities in Tebessa, Algeria, focusing on antimicrobial resistance (AMR) and phylogenetic relationships using whole-genome sequencing (WGS). The study employs SNP-based phylogenetic analysis to characterize strain relatedness and identifies key antibiotic resistance genes (ARGs) and sequence types (STs). The results suggest potential nosocomial transmission and high resistance rates to clinically relevant antibiotics, emphasizing a public health concern in this region. Before publication, several issues should be addressed to enhance clarity and completeness:
- The introduction cites the 2018 WHO priority pathogen list (Tacconelli et al.) but omits more recent global AMR burdendata (e.g., Murray, Christopher JL, et al. The Lancet, 2022, 399(10325): 629-655), which reports that AMR-associated UTIs caused 66,000 deaths in 2019, with North Africa among the high-burden regions. Integrating these figures would better contextualize the study’s focus on Algeria.
- The study collected isolates from only three healthcare facilities in Tebessa(lines 111 and 112). Please explain how representative these isolates are of Algeria’s broader UTI epidemiology and whether selection bias could affect generalizability.
- In line 62, the cited statistics on the global burden of UTIs are outdated. Update these figures with recent epidemiological data to reflect current trends in UTI incidence and mortality.
- In line 79, the term "CTX-M enzyme" should be fully spelled out upon its first mention.
- In line 79, please include a brief sentenceexplaining why CTX-M-producing strains limit clinical options.
Author Response
REVIEWER 1.
Comments and Suggestions for Authors
This manuscript presents a genomic epidemiology investigation of bacterial isolates
collected from healthcare facilities in Tebessa, Algeria, focusing on antimicrobial
resistance (AMR) and phylogenetic relationships using whole-genome sequencing
(WGS). The study employs SNP-based phylogenetic analysis to characterize strain
relatedness and identify key antibiotic resistance genes (ARGs) and sequence types
(STs). The results suggest potential nosocomial transmission and high resistance
rates to clinically relevant antibiotics, emphasizing a public health concern in this
region. Before publication, several issues should be addressed to enhance clarity
and completeness:
***Firstly, we want to thank the reviewer for their comments, which acknowledge
the interest of the study and help us improve the manuscript. All changes are
highlighted in yellow.
Please find below the point-by-point answer to the questions/corrections posed by
Reviewer 1.
1.1. The introduction cites the 2018 WHO priority pathogen list (Tacconelli et al.) but
omits more recent global AMR burden data (e.g., Murray, Christopher JL, et al. The
Lancet, 2022, 399(10325): 629-655), which reports that AMR-associated UTIs
caused 66,000 deaths in 2019, with North Africa among the high-burden regions.
Integrating these figures would better contextualize the study’s focus on Algeria.
***Thank you for the comment. In the revised version, we have integrated updated
information on the global burden of UTI, the impact of bacterial AMR, and specific
data on UTI in the Middle East and North Africa (MENA) region (Lines 65-73; 83-85;
110-115).
We have replaced 2018 WHO by the Medically Important Antimicrobial (MIA)
List, World Health Organization. 2024. 8 Feb. 2024,
https://www.who.int/philippines/news/detail-global/08-02-2024-who-medically-
important-antimicrobial-list-2024.
In addition, specific data on the occurrence of UTIs in Algeria and the implications
of E. coli and K. pneumoniae are discussed in the Discussion section (Lines 375-
379).
1.2. The study collected isolates from only three healthcare facilities in Tebessa
(lines 111 and 112). Please explain how representative these isolates are of Algeria’s
broader UTI epidemiology and whether selection bias could affect generalizability.
***Although our study was limited to three healthcare facilities in Tebessa, the
selected sites are key referral centers that serve a diverse patient population across
the province, making the sample reasonably representative of urban Algeria. The
Laboratory of Medical Analysis is a key reference center that receives a large and
diverse influx of patients from across the province, due to its advanced equipment
and strong reputation among local physicians who frequently refer to their patients
there. The Bouguerra Boulares Hospital and Khaldi Abdelaziz Maternity Hospital are
central healthcare institutions in the province, receiving patients from various
areas, particularly in complex or high-risk cases. These hospitals are well-equipped
and serve as referral centers for specialized care. Diagnostic approaches and
antimicrobial prescribing practices in these facilities mirror those used in other
regions of the country, where antibiotics remain widely accessible without
prescription, contributing to similar AMR dynamics. Additionally, our findings are
consistent with previous reports from northern and western Algeria, suggesting that
the observed AMR patterns in uropathogens may be reflective of national trends
(Nabti et al. 2019, DOI: 10.1089/mdr.2018.0314, in the north and Zenati et al. 2019,
DOI: 10.3855/jidc.10702, in the west).
In the new version of the manuscript, we have included a Limitation statement
based on the information mentioned above (please see the Discussion section,
Lines 389-397, and the Conclusions, Lines 730-740).
1.3. In line 62, the cited statistics on the global burden of UTIs are outdated. Update
these figures with recent epidemiological data to reflect current trends in UTI
incidence and mortality.
***We agree with the reviewer´s comment and in this new version we have updated
the requested information as follows: “Data from 1990 to 2021 revealed that UTIs
showed an upward trend, with a particularly pronounced disease burden among
women, older men, and low–middle socio-demographic index regions [5].
Furthermore, the impact estimation of UTI in 2021 was 287,200 deaths associated
with bacterial antimicrobial resistance (AMR), including 67,467 deaths directly
attributable to bacterial AMR, being the fourth leading cause of death associated
with AMR worldwide [He et al., 2025; Murray et al., 2022]” (Lines 65-71).
1.4.In line 79, the term "CTX-M enzyme" should be fully spelled out upon its first
mention.
***Thank you. Done (Line 96).
1.5.In line 79, please include a brief sentence explaining why CTX-M-producing
strains limit clinical options.
***We have integrated a brief explanation to clarify as follows “…cefotaximases
(CTX-M), a family of extended-spectrum beta-lactamase (ESBL) enzymes which
confer resistance to extended-spectrum cephalosporins (ESC), thereby limiting the
effectiveness of this last resort class of antibiotics and significantly reducing
available treatment options in clinical settings“ (Lines 96-100).
Reviewer 2 Report
Comments and Suggestions for Authors
This study “Phenotypic and Genotypic Characterization of ESBL-, AmpC-, and Carbapenemase-Producing Klebsiella pneumoniae and High-Risk Escherichia coli CC131, with the First Report of ST1193 as a Causative Agent of Urinary Tract Infections in Algeria” provides the phenotypic and genotypic profiles of several multidrug-resistant E. coli and K. pneumoniae isolates that cause urinary tract infections in a region of Algeria. This work's contribution is significant in this highly relevant topic in this country, as it could help understand the behavior of these bacteria and mitigate their spread, both locally and globally. However, certain aspects should be improved, such as specifying the sampling time period that remains to be determined, and organizing the findings in the discussion section in a more orderly manner. The focus could be solely on E. coli first, followed by the K. pneumoniae findings.
Specific comments
Line 26 “This study investigates the characteristics of these pathogens in the Tebessa region of Algeria”: State the objective of the study clearly and concisely; it is very general.
Line 27: mention the source of the isolates.
Line 99-101 “The present study comprised 57 non-duplicate isolates, 40 E. coli and 17 K. pneumoniae, recovered from urine samples at three healthcare facilities of north-east Algeria (Table S1 and Table S2)”: enter the period of time of collection
Line: 119 “…blaCARBA genes…”: Which carbapenemase genes were analyzed (VIM, IMP, OXA, NDM, KPC etc.)
Lines 137-143: place bibliographic reference
Line 150-151 “Presumptive identification of the pandemic clonal complex (CC)131 was performed via PCR screening for the rfbO25 and fliCH4 gene markers”: Provide a bibliographic reference on the use of these markers for clone identification.
Line 155-157 “Clonotyping identified 14 distinct fumC-fimH combinations. Notably, 42.5% of the isolates exhibited clonotype CH40-30, which is typically associated with CC131.”: Provide a bibliographic reference on the use of these markers for clone identification.
Line 162 “(Neg): Nine isolates showed no amplification of the 489-bp internal sequence [18].”: no amplification of the 489-bp internal sequence refers to the fact that it does not amplify the fimH gene?
Table 2: It is necessary to specify what the acronym NT means.
Line 254-258 “The phylogenetic tree was constructed with the CSI Phylogeny version 1.4 (CGE, https://cge.cbs.dtu.dk/ser-254 vices/CSIPhylogeny/ with the following parameters: min. depth at SNP positions ×10; min. relative depth at SNP positions: ×10; min. distance between SNPs (prune): 10 bp; min. SNP quality: 30; min. read mapping 256 quality: 25, a min. Z-score of 1.96 and by ignoring heterozygous SNPs). Branch values represent substitutions per site”: This paragraph should be placed in materials and methods
Line 278-279: “PCR analysis was conducted to detect the presence of blaESBL/AmpC/CARBA genes”: It is advisable to include which CARBA genes were identified?
Line 293-296 “The hypermucoviscous (HMV) phenotyping of the 17 K. pneumoniae isolates was assessed using the string test described by Shon et al. [23] with modifications on four different culture media: Mueller-Hinton (MH) agar, MacConkey (ML) agar, trypticase soy agar (TSA), and Columbia agar (5% sheep blood)”: This paragraph should be placed in materials and methods.
Overall Recommendation
Accept after Minor Revisions: The paper can in principle be accepted after revision based on the reviewer’s comments.
Author Response
REVIEWER 2
Comments and Suggestions for Authors
This study “Phenotypic and Genotypic Characterization of ESBL-, AmpC-, and
Carbapenemase-Producing Klebsiella pneumoniae and High-Risk Escherichia coli
CC131, with the First Report of ST1193 as a Causative Agent of Urinary Tract
Infections in Algeria” provides the phenotypic and genotypic profiles of several
multidrug-resistant E. coli and K. pneumoniae isolates that cause urinary tract
infections in a region of Algeria.
2.1. This work's contribution is significant in this highly relevant topic in this country,
as it could help understand the behavior of these bacteria and mitigate their spread,
both locally and globally.
However, certain aspects should be improved, such as specifying the sampling time
period that remains to be determined, and organizing the findings in the discussion
section in a more orderly manner. The focus could be solely on E. coli first, followed
by the K. pneumoniae findings.
***We want to thank the reviewer for their comments, which acknowledge the
interest and provide valuable feedback to improve the manuscript. All changes are
highlighted in yellow.
Please find below the point-by-point answer to the questions/corrections posed by
Reviewer 2.
Concerning the sampling period, it was already mentioned that the isolates were
collected between July 2022 and January 2024 in Materials and Methods (Lines 590-
591). In addition, the revised version includes this information in the Abstract
section (Line 30) and the Results section (Line 154).
Concerning the discussion organization, it was already shown by species. The first
part discusses E. coli (Lines 398-505) and the second part discusses K. pneumoniae
(Lines 506-556).
Specific comments
2.2. Line 26 “This study investigates the characteristics of these pathogens in the
Tebessa region of Algeria”: State the objective of the study clearly and concisely; it
is very general.
***Thank you for the comment. In response to reviewers' comments (specifically 2
and 3), we have streamlined the Results section of the abstract, refined the
objective description, and specified the source of the isolates. Furthermore, we
have emphasized the clinical implications of our findings and proposed a
surveillance strategy in the revised version.
Concerning the aim of the study, the revised sentence now reads: “This study aimed
to provide an overview of the virulence factors, clonal diversity, and antibiotic
resistance profiles of extended-spectrum cephalosporin (ESC)-E. coli and K.
pneumoniae causing UTIs in humans in the Tebessa region of Algeria” (Lines 26-29).
In addition, “a simplified method based on virulence traits for E. coli and K.
pneumoniae is proposed here for antimicrobial resistance (AMR) monitoring” (Lines
52-54).
2.3. Line 27: mention the source of the isolates.
***We have included the source of the isolates. The revised sentence now reads:
“Forty E. coli and 17 K. pneumoniae isolates exhibiting ESC-resistance were
recovered (July 2022-January 2024) from urine samples of patients at three
healthcare facilities to be phenotypically and genotypically characterized.” (Lines
29-32).
2.4. Line 99-101 “The present study comprised 57 non-duplicate isolates, 40 E. coli
and 17 K. pneumoniae, recovered from urine samples at three healthcare facilities
of north-east Algeria (Table S1 and Table S2)”: enter the period of time of collection.
***Thank you for the comment. The revised version includes this information in the
Abstract section (Line 30) and the Results section (Line 154) in addition to Materials
and Methods (Lines 590-591).
2.5. Line: 119 “…blaCARBA genes…”: Which carbapenemase genes were analyzed
(VIM, IMP, OXA, NDM, KPC etc.)
*** Thanks for this observation. As shown in the Materials and Methods section of
the manuscript, carbapenemase genes were detected by PCR using specific
primers targeting the most clinically relevant families: VIM, IMP, OXA, NDM, KPC. To
enhance clarity, we have now explicitly stated this information in the main text, and
the revised sentence reads as follows: “The isolates were subsequently analyzed by
PCR to detect the presence of blaESBL (SHV, CTX-M), blaCMY, and blaCARBA (VIM, IMP,
OXA, NDM, KPC) genes, followed by Sanger sequencing” (Lines 173-175).
2.6. Lines 137-143: place bibliographic reference
***Thank you for the comment. A bibliographic reference has now been added to
support the information presented in this section (Line 193).
2.7. Line 150-151 “Presumptive identification of the pandemic clonal complex
(CC)131 was performed via PCR screening for the rfbO25 and fliCH4 gene markers”:
Provide a bibliographic reference on the use of these markers for clone
identification.
***Thank you for the comment. Bibliographic references have now been added to
support the information presented in this section (Lines 206 and 208).
2.8. Line 155-157 “Clonotyping identified 14 distinct fumC-fimH combinations.
Notably, 42.5% of the isolates exhibited clonotype CH40-30, which is typically
associated with CC131.”: Provide a bibliographic reference on the use of these
markers for clone identification.
***Three bibliographic references have now been added to support the information
presented in this section (Line 212).
2.9. Line 162 “(Neg): Nine isolates showed no amplification of the 489-bp internal
sequence [18].”: no amplification of the 489-bp internal sequence refers to the fact
that it does not amplify the fimH gene?
***A negative amplification of the 489-bp internal sequence does not mean the
strain is negative for the fimH gene carriage. The nine isolates were, in fact, positive
for fimH screening, which was detected using the primers and conditions described
in Johnson and Stell, 2000 (DOI: 10.1086/315217). However, the primers targeting
the 489-bp internal fragment did not yield a positive result. This suggests sequence
polymorphisms within the targeted region affecting typing amplification, or other
genetic events within the gene.
To clarify this, the following explanation is included as foot note: “(Neg): Nine
isolates showed no amplification of the 489-bp internal sequence of the fimH gene
targeted for clonotyping [Weissman et al., 2012] despite all 40 E. coli isolates testing
positive for fimH detection using primers and conditions described by Johnson and
Stell [2000].” (Lines 217-219).
2.10. Table 2: It is necessary to specify what the acronym NT means.
***Thank you for the comment. It is now included in the table legend “NT, Not
Typable.” (Line 248).
2.11. Line 254-258 “The phylogenetic tree was constructed with the CSI Phylogeny
version 1.4 (CGE, https://cge.cbs.dtu.dk/ser-254 vices/CSIPhylogeny/ with the
following parameters: min. depth at SNP positions ×10; min. relative depth at SNP
positions: ×10; min. distance between SNPs (prune): 10 bp; min. SNP quality: 30;
min. read mapping 256 quality: 25, a min. Z-score of 1.96 and by ignoring
heterozygous SNPs). Branch values represent substitutions per site”: This
paragraph should be placed in materials and methods
***Thank you for the comment. It is now integrated in Materials and Methods (Lines
703-708).
2.12. Line 278-279: “PCR analysis was conducted to detect the presence of
blaESBL/AmpC/CARBA genes”: It is advisable to include which CARBA genes were
identified?
***Thank you for the comment. We have now included this information in the main
text; the revised sentence reads as follows: “PCR analysis was conducted to detect
the presence of blaESBL (SHV, CTX-M), blaCMY, and blaCARBA (VIM, IMP, OXA, NDM, KPC)
genes, followed by Sanger sequencing.”(Lines 334-336).
2.13. Line 293-296 “The hypermucoviscous (HMV) phenotyping of the 17 K.
pneumoniae isolates was assessed using the string test described by Shon et al.
[23] with modifications on four different culture media: Mueller-Hinton (MH) agar,
MacConkey (ML) agar, trypticase soy agar (TSA), and Columbia agar (5% sheep
blood)”: This paragraph should be placed in materials and methods.
***Thank you for the comment. As this information is already described in the
Materials and Methods section, we have removed it to avoid redundancy. Now the
text in the Results section reads as follows: “The hypermucoviscous (HMV)
phenotyping of the 17 K. pneumoniae isolates, using the string test described by
Shon et al. [23] with modifications, showed that four were classified as HMV-positive
in at least two of the tested culture conditions (Table 4 and S2). “ (Lines 350-353).
Overall Recommendation
Accept after Minor Revisions: The paper can in principle be accepted after revision
based on the reviewer’s comments.
***We would like to sincerely thank the reviewer for their positive evaluation and
constructive feedback throughout the revision process. We appreciate the
recommendation for acceptance and have addressed all the minor revisions
accordingly.
Reviewer 3 Report
Comments and Suggestions for Authors
Comments to manuscript: antibiotics-3575826:
Phenotypic and Genotypic Characterization of ESBL, AmpC, and Carbapenemase-Producing Klebsiella pneumoniae and High-Risk Escherichia coli CC131, with the First Report of ST1193 as a Causative Agent of Urinary Tract Infections in Algeria
Abstract:
Abstract is well written and need no corrections. However, if authors find it comfortable to reduce the length of Results part it will be of much brief and optimum word counts. Please also consider adding a sentence or two to the abstract that highlights the clinical implications of your findings.
Introduction:
Line 57-73: Only three -four references are used here, of which two are used repeatedly. Use and cite different references here to increase/widen the literature consulted while conducting the research. It will improve the scope of the study.
Line 58: Urinary tract infections (UTIs) are among the most common infectious diseases worldwide…….Please add in Human. As the journal also publish a large number of veterinary papers please try to include Human word wherever necessary. If possible please also add it in Title of the manuscript.
The introduction however, is well written, it could benefit from a slightly expanded inputs on the specific challenges related to antimicrobial resistance in Algeria and the Tebessa region. Consider adding a sentence or two about the local healthcare setting and common antibiotic prescription practices.
The emergence of ST1193 is noted, but its clinical significance and the reasons for considering it an "emerging clone" could be elaborated on.
Results:
The results are generally well-organized. The use of tables and figures is very effective.
Figure 1: Consider adding a brief description of Figure 1 in the text, highlighting the key findings (e.g., "Figure 1 shows the high prevalence of resistance to…").
Table 1: The table is clear, but the abbreviations "UPEC status," "ExPEC status," and "FQR" could be spelled out in the caption for clarity, even though they are defined later.
Discussion
Limitations: It would strengthen the manuscript to acknowledge any limitations of the study (e.g., sample size, geographical representation, specific methods used).
Implications: The discussion could be expanded to further explore the implications of the findings for clinical practice, infection control, and public health policy in Algeria. Consider suggesting specific interventions or strategies that could be implemented to address the spread of MDR uropathogens.
Conclusion:
Line 627: Use a period/full stop punctuation at end.
General Comments:
This is a well-conducted and clearly presented study that contributes valuable data to the understanding of antimicrobial resistance in UTIs in Algeria. By addressing the specific comments and suggestions above, the authors can further enhance the quality and impact of their manuscript.
Author Response
REVIEWER 3
Comments and Suggestions for Authors
Comments to manuscript: antibiotics-3575826:
Phenotypic and Genotypic Characterization of ESBL, AmpC, and Carbapenemase-
Producing Klebsiella pneumoniae and High-Risk Escherichia coli CC131, with the
First Report of ST1193 as a Causative Agent of Urinary Tract Infections in Algeria
Abstract:
3.1. Abstract is well written and need no corrections. However, if authors find it
comfortable to reduce the length of Results part it will be of much brief and optimum
word counts. Please also consider adding a sentence or two to the abstract that
highlights the clinical implications of your findings.
***We would like to sincerely thank the reviewer for their positive evaluation and
constructive feedback. Taking into consideration reviewers´ 3 and 2 comments, we
have reduced the Results, improved the objective description, and mentioned the
source of the isolates. We have also highlighted the clinical implications of the
findings and a proposal for surveillance (Lines 23-54).
Introduction:
3.2. Line 57-73: Only three -four references are used here, of which two are used
repeatedly. Use and cite different references here to increase/widen the literature
consulted while conducting the research. It will improve the scope of the study.
***Thank you for the suggestion and opportunity to improve the introduction. We
have expanded the introduction based on more than 20 references. Considered
Reviewer´s 1 comments together with yours, we have integrated updated
information on the global burden of UTI, the impact of bacterial AMR, and specific
data on UTI in the Middle East and North Africa (MENA) region (Lines 65-73; 83-85;
110-115). We have replaced 2018 WHO by the Medically Important Antimicrobial
(MIA) List, World Health Organization 2024, and we have included more information
on specific reports in Algeria (Lines 116-142).
3.3. Line 58: Urinary tract infections (UTIs) are among the most common infectious
diseases worldwide…….Please add in Human. As the journal also publish a large
number of veterinary papers please try to include Human word wherever necessary.
If possible please also add it in Title of the manuscript.
***Thank you for the suggestion. The title of the new version has been modified to
include “Human” as follows “…Urinary Tract Infections in Human patients in
Algeria”. It has also been specified whenever necessary throughout the text.
3.4. The introduction however, is well written, it could benefit from a slightly
expanded inputs on the specific challenges related to antimicrobial resistance in
Algeria and the Tebessa region. Consider adding a sentence or two about the local
healthcare setting and common antibiotic prescription practices.
***Thank you for your valuable suggestion. We have expanded the Introduction
(Lines 135-142) to include specific information on the challenges related to
antimicrobial resistance in Algeria, such as the correlation between inappropriate
antibiotic use and the rising prevalence of ESBL-producing bacteria. This includes
reference to the lack of proper medical oversight, over-the-counter access to
antibiotics, and the limited availability of coordinated national surveillance data.
Additionally, we have incorporated further contextual details in the Discussion
section (Lines 557–579) to reinforce the relevance of our study within the local
healthcare landscape.
3.5. The emergence of ST1193 is noted, but its clinical significance and the reasons
for considering it an "emerging clone" could be elaborated on.
***Thank you for this insightful comment. We have now elaborated on the clinical
significance of E. coli ST1193 in the revised manuscript. As noted by Pitout et al.
(2022) [Reference 11], ST1193 is considered an emerging, globally disseminated
multidrug-resistant clone, increasingly associated with urinary tract and
bloodstream infections. It shares several characteristics with the pandemic ST131
lineage, including fluoroquinolone resistance, the ability to acquire extended-
spectrum β-lactamase (ESBL) genes, and enhanced virulence traits such as
increased biofilm formation and pathogenicity. Its rapid international spread and
association with community and hospital infections highlight its public health
importance (Lines 89-95).
Results:
3.6. The results are generally well-organized. The use of tables and figures is very
effective.
***Thank you very much for your positive feedback.
3.7. Figure 1: Consider adding a brief description of Figure 1 in the text, highlighting
the key findings (e.g., "Figure 1 shows the high prevalence of resistance to…").
***Modified as follows “ As shown in Figure 1, most of the isolates (≥ 50%) exhibited
resistance to aztreonam, amoxicillin-clavulanic acid, nalidixic acid,
sulfamethoxazole, ciprofloxacin, doxycycline, and tobramycin. Noticeably, four
isolates (10%) were resistant to ertapenem. Overall, all isolates were classified as
MDR, exhibiting resistance to at least one drug in three or more antimicrobial
categories [23], and 62.5% demonstrated FQR (Figure 1)” (Lines 166-172).
3.8. Table 1: The table is clear, but the abbreviations "UPEC status," "ExPEC status,"
and "FQR" could be spelled out in the caption for clarity, even though they are
defined later.
***Done as suggested (Lines 219-221).
Discussion
3.9. Limitations: It would strengthen the manuscript to acknowledge any limitations
of the study (e.g., sample size, geographical representation, specific methods
used).
***Thank you for the suggestion We have included a paragraph acknowledging the
limitations of the study which reads as follows: “This study had certain limitations,
as the isolates were collected from only three healthcare facilities within a single
province (Tebessa), which may limit the broader geographical representation of the
findings. Additionally, the sample size, while sufficient for exploratory analysis, may
not fully capture the genetic and phenotypic diversity of E. coli and K. pneumoniae
circulating in other regions of Algeria. Nevertheless, our findings are consistent with
previous reports from northern and western Algeria, suggesting that the AMR
patterns observed in uropathogens may reflect broader national trends.
Furthermore, future studies involving in-depth genomic characterization, such as
plasmid profiling and mobility analyses, are needed to better under-stand the
dynamics of resistance gene dissemination.” (Lines 730-740).
3.10. Implications: The discussion could be expanded to further explore the
implications of the findings for clinical practice, infection control, and public health
policy in Algeria. Consider suggesting specific interventions or strategies that could
be implemented to address the spread of MDR uropathogens.
***Thank you for the suggestion. In Algeria, the widespread and often unregulated
use of antibiotics, particularly β-lactams, has been identified as a major factor
contributing to the emergence and spread of ESBL-producing isolates. A recent
statistical study conducted in Tebessa province by (Fares et al., 2023) further
emphasizes this issue, highlighting the correlation between antibiotic misuse and
the increasing prevalence of resistant pathogens. Similarly, (Nabti et al., 2019)
highlighted a high prevalence of ESBL-producing E. coli in northeastern Algeria,
attributing it to inappropriate antibiotic prescriptions. In the northwestern region,
(Zenati et al., 2019) found that the high rate of resistance at a university-affiliated
hospital in Tlemcen was associated with the indiscriminate sale of antibiotics,
including over-the-counter access without medical supervision. Collectively, these
findings underscore the urgent need for comprehensive antibiotic stewardship
programs and the implementation of strict national policies to regulate antibiotic
use across all regions.
We would also like to emphasize that data on antimicrobial resistance in Algeria
remain limited. As noted by (Touati & Mairi, 2020), most findings published to date
originate from isolated academic research, particularly doctoral studies conducted
in collaboration with European laboratories, rather than from national surveillance
efforts. This highlights the importance of region-specific studies like ours to fill
critical data gaps and inform future health policies in Algeria.
The Algerian Antimicrobial Resistance Network (AARN) reported in its annual
evaluation that the antimicrobial resistance (AMR) situation concerning Escherichia
coli and Klebsiella pneumoniae in healthcare settings across Algeria is becoming
increasingly alarming. The recurrent isolation of resistant strains, particularly in
hospital environments, highlights the growing threat these pathogens pose to public
health. The network's findings, which are based on data collected from nearly all
university hospital centers nationwide, offer a comprehensive overview of the
current AMR landscape in Algeria. These trends underscore significant challenges
within the healthcare system, particularly with respect to complex resistance
patterns and the disparities in diagnostic capabilities among laboratories.
The above information has been included in the Discussion section (Lines 557-580).
Conclusion:
3.11. Line 627: Use a period/full stop punctuation at end.
***Thank you. Done.
General Comments:
This is a well-conducted and clearly presented study that contributes valuable data
to the understanding of antimicrobial resistance in UTIs in Algeria. By addressing the
specific comments and suggestions above, the authors can further enhance the
quality and impact of their manuscript.
***We thank the reviewer for their kind and encouraging feedback. We are pleased
that the study was found to be well-conducted and clearly presented, and we
appreciate the recognition of its contribution to understanding antimicrobial
resistance in UTIs in Algeria. We have carefully addressed all specific comments
and suggestions to further improve the quality and impact of the manuscript.
Reviewer 4 Report
Comments and Suggestions for Authors
This study aimed to investigate the molecular characteristics of E. coli and K. pneumoniae exhibiting ESC resistance associated with UTIs in Algeria where data on such infections remains scarce. The comments are given below-
- Please mention the library kit used in this study.
- Please provide the URL and references for ResFinder 4.6., VirulenceFinder 2.0, PlasmidFinder 2.1/pMLST 2.0, CHTyper 1.0, and SeroTypeFinde.r
- In Table S1, virulence gene results are not detected for all isolates such as EC1a, Ec37 etc. Why?
- In Table S2 and in line number 119-121: without doing WGS or sanger sequencing, how could authors identify the variants of ESBLs or AmpC such as CTX-M-15, CTX-M-194, CTX-M-27, SHV-148, CMY66/159 etc?
- Authors are advised to do sanger sequencing to detect the variants for all the isolates especially for ESBL and AmpC genes.
- Line 116: “Overall, all isolates were classified as MDR”----------add the criteria of a MDR strains with proper reference.
- Due to the lack of studies regarding the high-risk clones of coli from a low- and middle-income country such as Algeria, authors are advised not only to detect the AMR genes by PCR analysis, rather to do the WGS of 17 CC131 E coli high-risk isolates.
- They could analyse if there any differences are present in the surrounding region of the blaCTX-M gene as all 17 isolates possessed this gene. Moreover, they could also try to identify if any differences present between the UPEC and ExPEC isolates among the high-risk group.
- A core genome analysis needs to be done to compare these 17 strains with globally available CC131 strains harbouring CTX-M and/or AmpC from different countries to compare the strains with respect to resistance, plasmid incompatibility groups and virulence. Include a figure for this analysis. Discussed the interpretation of the figure in the result portion.
- Moreover, authors are also suggested to interpret the location of ESBL and AmpC genes either on plasmid or chromosome from the WGS data and should validate the results with conjugation or transformation assays.
- Figure 1, 3 could not be included in the main manuscript. Authors can upload them as supplementary files.
- Why did authors include only five ST1193 isolates from a single country to compare them with the studied strain of ST1193? How many genomes of ST1193 are actually available in NCBI? I think there is no significance of making Figure 2A and 2B. Authors can make a single comparative figure including available global ST1193 isolates and the studied strain.
- Quality of figure 2 is very poor. Results were interpreted from this figure very poorly. Authors must represent this comparative analysis with iTOL representation considering different resistance genes detected in the studied strain and global isolates. In addition, source of isolation, year of isolation, and virulence genes need to be discussed and include all analysis in the result portion of the manuscript.
- Table 2: include the resistance genes, and plasmid incompatibility group.
- Authors identified OXA-48 genes in three E. coli isolates, two of which possessed CTX-M-27. Again, raise the same question that without doing WGS or sanger sequencing how could authors denote CTX-M as CTX-M-27 or OXA-48-like as OXA-48? They might be other variants also. Please do WGS to confirm the variants of OXA-48-like and CTX-M at least for these 2 E. coli isolates harbouring OXA-48-like and CTX-M-27 as authors claimed that this is the first report of OXA-48 harbouring E. coli co-harbouring CTX-M-27 from Algeria.
- Discuss the gene surrounding (OXA-48, CTX-M-27) of these 2 isolates, their ST type, mobile genetic elements etc. Also check the transmissibility of the genes.
- Two isolates of K. pneumoniae were detected as hypervirulent harbouring NDM. As per WHO 2024, carbapenem-resistant K. pneumoniae is considered as the most critical organism. In case of carbapenem-resistant hypervirulent KP, the case is more crucial. Authors are advised to do WGS for these 2 strains to reveal high inter-genomic resemblance with the other hvKP reference genomes and they are also advised to check the transfer of plasmid-associated genes (peg-465 344, rmpA, and iucA) along with blaNDM. Also detect whether these 2 strains belonged to any high-risk clone or not.
- As per the revised work suggested here, please discuss all the sections in the discussion part of the revised manuscript.
English language quality could be improved.
Author Response
REVIEWER 4
Comment
The title of the paper emphasizes “Phenotypic and Genotypic Characterization of
ESBL-, AmpC-, and Carbapenemase-Producing Klebsiella pneumoniae and High-
Risk Escherichia coli CC131, with the First Report of ST1193 as a Causative Agent of
Urinary Tract Infections in Algeria”.
Without doing any sequencing (at least Sanger sequencing), based on only PCR,
how could the authors identify the antimicrobial resistance gene variants? In case
of determining the UPEC and ExPEC E. coli or screening of high-risk E. coli in UTIs
based on previously established articles are satisfactory but this was not the sole
aim of this study as much as I could understand from the manuscript. Don’t need to
do WGS, but without sanger seq it is not possible to characterize the genes.
***We appreciate the reviewer’s engagement with our work. However, we
respectfully clarify that the manuscript does not rely solely on PCR screening for
variant-level identification. As detailed in the Materials and Methods section (Lines
629–632; Line 648; Supplementary Tables S6 and S10), Sanger sequencing was
performed to characterize bla gene variants and to determine fumC and fimH alleles
used for clonotyping. Furthermore, whole genome sequencing (WGS) was
conducted for the ST1193 isolate (LREC-468), enabling in-depth in silico analysis of
resistance genes, mutations, plasmid content, virulence determinants, and
phylogenetic context (Lines 253-319; 674–714, Table 3, Figure 2).
We have clarified this point in the revised manuscript to avoid any misunderstanding
regarding the methodological rigor and scope of our molecular characterization
(Discussion, Lines 557–579).
The isolates were analysed by PCR to detect the presence of blaESBL (SHV, CTX-M),
blaCMY, and blaCARBA (VIM, IMP, OXA, NDM, KPC) genes, followed by Sanger
sequencing.
My revised comments are-
1. Please mention the library kit used in this study.
*** We thank the reviewer for this comment. As requested, we have now included
the specific library preparation kit used for sequencing. The revised text in the
Materials and Methods section now reads: “...following library preparation using the
standard Illumina DNA Prep kit.” (Lines 682-683).
2. Please provide the URL and references for ResFinder 4.6.,
VirulenceFinder 2.0, PlasmidFinder 2.1/pMLST 2.0, CHTyper 1.0, and
SeroTypeFinde.r
** We thank the reviewer for this comment. As requested, we have now included
comprehensive information concerning references and URLs of all the CGE tools
used here, as follows (Lines 688-698).
“The assembled contigs, with genomic size 5.0 Mbp, were analyzed using the
bioinformatics tools of the Center for Genomic Epidemiology (CGE) as specified,
and applying the thresholds suggested by default when required (minimum identity
of 95% and coverage of 60%): for the presence of acquired genes and or
chromosomal mutations mediating antimicrobial resistance (ResFinder 4.6.) [91,
92]; for identification of acquired virulence genes (VirulenceFinder 2.0) [93, 94];
plasmid replicon types (PlasmidFinder 2.1/pMLST 2.0) [95]; identification of
clonotypes (CHTyper 1.0) [96]; and serotypes (SeroTypeFinder 2.0.1) [97]. Two
different MLST (2.0.9) schemes were applied for phylogenetic typing [34, 98].
Additionally, cgMLSTFinder1.2 was called for the core genome multi-locus typing
(cgMLST) [92, 99].”
3. In Table S1, virulence gene results are not detected for all isolates such
as EC1a, Ec37 etc. Why?
***We thank the reviewer for this observation. As indicated in the Materials and
Methods section, the virotype classification was only applied to O25b:H4-B2-
CH40-30 isolates belonging to CC131, in accordance with the protocol established
by Dahbi et al. [30]. This virotyping scheme is specific to ST131 and is not applicable
to other clonal lineages, which is why virulence gene results are not shown for
isolates outside this clonal complex (e.g., EC1a, EC37).
4. In Table S2 and in line number 119-121: without doing sanger sequencing,
how could authors identify the variants of ESBLs or AmpC such as CTX-
M-15, CTX-M-194, CTX-M-27, SHV-148, CMY66/159 etc?
***We appreciate the reviewer’s attention to this aspect. As described in the
Materials and Methods section and indicated in Table S6, Sanger sequencing was
indeed performed for PCR products obtained from ESBL and AmpC gene
amplification, in order to identify specific variants. We have now revised the text in
the Materials and Methods and Results sections to clearly indicate that
sequencing was carried out to confirm gene variants or subtypes. Specifically, in
Lines 173-175 “The isolates were subsequently analyzed by PCR to detect the
presence of blaESBL (SHV, CTX-M), blaCMY, and blaCARBA (VIM, IMP, OXA, NDM, KPC)
genes, followed by Sanger sequencing.”
Lines 334-336 “PCR analysis was conducted to detect the presence of blaESBL (SHV,
CTX-M), blaCMY, and blaCARBA (VIM, IMP, OXA, NDM, KPC) genes, followed by Sanger
sequencing.”
5. Authors are advised to do sanger sequencing to detect the variants for all
the isolates especially for ESBL and AmpC genes.
***We appreciate the reviewer’s attention to this aspect. As described in the
Materials and Methods section and indicated in Table S6, Sanger sequencing was
indeed performed for PCR products obtained from ESBL and AmpC gene
amplification, in order to identify specific variants. We have now revised the text in
the Materials and Methods and Results sections to clearly indicate that
sequencing was carried out to confirm gene variants or subtypes. Specifically, in
Lines 173-175 “The isolates were subsequently analyzed by PCR to detect the
presence of blaESBL (SHV, CTX-M), blaCMY, and blaCARBA (VIM, IMP, OXA, NDM, KPC)
genes, followed by Sanger sequencing.”
Lines 334-336 “PCR analysis was conducted to detect the presence of blaESBL (SHV,
CTX-M), blaCMY, and blaCARBA (VIM, IMP, OXA, NDM, KPC) genes, followed by Sanger
sequencing.”
6. Line 116: “Overall, all isolates were classified as MDR”----------add the
criteria of a MDR strains with proper reference.
***We appreciate the reviewer’s attention to this aspect. As described in the
Materials and Methods section “Isolates were classified as MDR if they exhibited
resistance to at least one drug in three or more antimicrobial categories [Magiorakos
et al., 2012, Reference 23] (Lines 625-627).
We have now included this criterion in Lines 170-172 and 332-333 of the Results for
clarification.
7. Check the transmissibility of the gene to assure their presence on
plasmid.
***We thank the reviewer for this suggestion. However, we respectfully note that
the aim of the present study is to characterize the antibiotic resistance and
virulence profiles of E. coli and K. pneumoniae isolates causing UTIs in the
Tébessa region of Algeria, within the context of a streamlined and cost-effective
laboratory workflow for AMR monitoring. As outlined in the Introduction and
Discussion, the primary objective was to generate relevant local data to support
surveillance efforts in low- and middle-income settings, where resource-
intensive approaches such as conjugation assays or full plasmid sequencing are
often not feasible.
This study builds on previously validated workflows (doi:
10.3389/fcimb.2024.1351618; doi: 10.1128/spectrum.00041-22) and
demonstrates that rapid screening based on virulence and resistance traits is a
viable and scalable approach for early detection and outbreak monitoring in
resource-limited settings.
Extending the study to include conjugation experiments or in-depth plasmid
analysis would fall outside the scope and objectives of the current work and
would require significant additional time and resources. Importantly, this project
was conducted as part of a doctoral training initiative, enabling practical and
locally relevant research with direct public health value.
We would also like to highlight that other reviewers acknowledged the clarity,
scientific merit, and utility of our approach, with Reviewer 3 noting that: “This is a
well-conducted and clearly presented study that contributes valuable data to the
understanding of antimicrobial resistance in UTIs in Algeria.”
Finally, we have now included an explicit statement in the Discussion regarding the
study’s limitations and the need for future in-depth genomic characterization
(e.g., plasmid profiling, mobility studies) to better understand the dynamics of
resistance gene dissemination (Lines 730-740).
8. Figure 1, 3 could not be included in the main manuscript. Authors can
upload them as supplementary files.
***We respectfully disagree with the suggestion to move Figures 1 and 3 to the
supplementary material. These figures enhance the readability and Reviewer 3
explicitly noted that “the use of tables and figures is very effective”.
However, we remain fully open to aligning with the editorial decision, should the
journal recommend relocating them.
9. Why did authors include only five ST1193 isolates from a single country
to compare them with the studied strain of ST1193? How many genomes
of ST1193 are actually available in NCBI? I think there is no significance
of making Figure 2A and 2B. Authors can make a single comparative figure
including available global ST1193 isolates and the studied strain.
***We thank the reviewer for this comment. Our intention was not to provide a
comprehensive phylogeographic reconstruction of ST1193, but to support the
novelty of our finding—the first report of an ST1193 isolate in Algeria causing a
UTI with a resistance and virulence profile consistent with global high-risk
clones.
To justify our approach:
• cgST criteria: We used core genome sequence typing (cgST) to identify
highly similar ST1193 isolates, as described by Achtman et al. (2022)
(https://doi.org/10.1098/rstb.2021.0240 ). According to this classification,
the only publicly available genome assigned to cgST140226 (the cgST of
our isolate) in EnteroBase was ESC_RA5887AA from the University of Oxford.
• Comparative strategy: We compared this isolate and five ST1193 genomes
we previously sequenced from Spain (as described in García-Meniño et al.,
2022), all of which belong to the same clonal background. The SNP-based
phylogeny was limited to closely related genomes to demonstrate the
genetic similarity within this clone and to confirm the genomic congruence
of the Algerian ST1193 isolate with those reported globally.
• On NCBI ST1193 genome availability: While hundreds of ST1193 genomes
exist in public repositories such as NCBI, very few are annotated with
cgMLST identifiers, and most are not suited for direct SNP comparison due
to fragmented assemblies or missing metadata. Our analysis focused on
quality-filtered, closely comparable genomes with relevant metadata.
• Relevance of Figure 2: Figures 2A and 2B support the primary goal of our
study—to report and place this first Algerian ST1193 isolate in a genomic
context, showing its relatedness to both a publicly deposited genome and
our Spanish clinical isolates. This provides sufficient resolution to show it
belongs to the emerging, globally disseminated clone and to highlight the
public health relevance of this finding.
Therefore, we respectfully believe that replacing these figures with a large-scale
tree based on globally diverse ST1193 isolates would not be aligned with the scope
of our work, which focuses on detection and initial characterization, not global
evolutionary analysis.
10. Quality of figure 2 is very poor. Results were interpreted from this figure
very poorly. Authors must represent this comparative analysis with iTOL
representation considering different resistance genes detected in the
studied strain and global isolates.
***We appreciate the reviewer’s comment regarding Figure 2. However, we would
like to respectfully maintain the current format for several reasons:
• CGE tool compliance: The figure was generated using the CSI Phylogeny
tool from the Center for Genomic Epidemiology (CGE), which ensures
standardized SNP-based phylogeny, clarity, and compliance with journal
technical specifications.
• Reviewer 3’s appreciation: As noted by Reviewer 3, the figures and tables
were clear and effective in supporting the manuscript's conclusions. We
are hesitant to modify a figure that was positively received and aligns with
our paper’s focus.
• Clarity of analysis: Figure 2 clearly visualizes the SNP distances between
the Algerian ST1193 and its closest relatives, supporting the textual
interpretation that this clone shares high genomic similarity with other
internationally circulating ST1193 strains.
• Scope limitations: Rebuilding the tree in iTOL using broader data, and
adding ARGs for each isolate, would require curating a comprehensive set of
globally distributed ST1193 isolates with comparable sequencing depth and
annotation, which falls beyond our scope. Our primary goal is to confirm that
our Algerian isolate shares resistance and virulence profiles consistent
with global ST1193 trends, as already illustrated through ResFinder,
VirulenceFinder, and plasmid content (Table 3).
11. Table 2: include the resistance genes, and plasmid incompatibility group.
*** The acquired resistance genes and plasmid incompatibility groups are already
included in Table 3 for the ST1193 isolate, which was the one WGS. Table 2
summarizes phenotypic resistance and virulence profiles (virotype scheme) of
CC131 isolates. However, to meet the reviewer’s request, we have added a column
with the resistance genes determined by PCR and Sanger sequencing.
12. Authors identified OXA-48 genes in three E. coli isolates, two of which
possessed CTX-M-27. Again, raise the same question that without doing
sanger sequencing how could authors denote CTX-M as CTX-M-27 or
OXA-48-like as OXA-48? They might be other variants also. Please do
sanger sequencing to confirm the variants of OXA-48-like and CTX-M at
least for these 2 E. coli isolates harbouring OXA-48-like and CTX-M-27 as
authors claimed that this is the first report of OXA-48 harbouring E. coli
co-harbouring CTX-M-27 from Algeria.
***We appreciate the reviewer’s attention to this aspect. Please, see answers to
comments 4, 5 and 7.
13. Also check the transmissibility of the genes.
***We thank the reviewer for this suggestion. However, we respectfully note that
the aim of the present study is of a different nature and aim (please see also answer
7). In this new version, we have globally addressed the comments and requirements
from all reviewers (four). Furthermore, we have included a Limitation statement for
further clarification (Lines 730-740).
14. Two isolates of K. pneumoniae were detected as hypervirulent
harbouring NDM. Check the variant of NDM by sanger sequencing.
***Please, visit answers to comments 4, 5 and 7. We have now included an explicit
statement in the Discussion regarding the study’s limitations and the need for
future in-depth genomic characterization (e.g., plasmid profiling, mobility studies)
to better understand the dynamics of resistance gene dissemination
15. As per the revised work suggested here, please discuss all the sections in
the discussion part of the revised manuscript.
***We thank the reviewer for this suggestion. The Discussion section has been
thoroughly revised and expanded in response to the comments from all reviewers,
all changes have been highlighted in yellow.
Comments on the Quality of English Language
English language quality could be improved.
***We thank the reviewer for this observation. The manuscript has been carefully
reviewed and edited for English language, grammar, and style. We have improved
sentence structure and clarity throughout the manuscript to enhance its
readability.
Round 2
Reviewer 4 Report
Comments and Suggestions for Authors
CTXM-15 is the predominant gene detected in almost all E. coli and KP strains. They have written their manuscript’s title as Genotypic Characterization of ESBL-, AmpC-, and Carbapenemase-Producing Klebsiella pneumoniae and High-Risk Escherichia coli CC131-----“. However, only documenting presence/absence of different resistance genes could not be claimed as Genotypic Characterization. Authors still disagree to check the transmissibility of the genes by basic experiment such as conjugation or to check the incompatibility groups of the plasmids carrying those resistance genes by PCR-based method or genetic environment of the ESBL gene or carbapenemase genes which are the minimum important data to characterize the ESBL or carbapenemase-producing strains. This could not be kept under limitation of the paper.
Moreover, the authors claim that “Our intention was not to provide a
comprehensive phylogeographic reconstruction of ST1193, but to support the
novelty of our finding—the first report of an ST1193 isolate in Algeria causing a
UTI with a resistance and virulence profile consistent with global high-risk
clones-----------a pictorial presentation is always more clear to understand the resistance and virulence profile consistency with global high-risk
clones and that’s why I suggested to use iTOL for more clarity. Representation of a tree in iTOL would not require broader data. If you are comparing between 6-7 isolates, then also one can add ARGs, virulence or any other profiles to the tree (as separate column) for more clarity as this is the first report from Algeria the ST1993 causing UTI. From this type of figure, it would be clear the resistance or virulence consistency between the strains. However, the authors did not discuss the prevalence of resistance genes or virulence genes in global ST1993 isolates (at least those included in the figure) in comparison to their own isolate and also disagree to modify the figure.
Author Response
"Please see the attachment"
